# Combined deletion of Glut1 and Glut3 impairs lung adenocarcinoma growth

Caroline Contat[1,2], Pierre-Benoit Ancey[1,2], Nadine Zangger[1,2,3], Silvia Sabatino[1,2], Justine Pascual[1,2], Stéphane Escrig[4], Louise Jensen[4], Christine Goepfert[5], Bernard Lanz[6], Mario Lepore[7], Rolf Gruetter[6,7], Anouk Rossier[1,2], Sabina Berezowska[8], Christina Neppl[8], Inti Zlobec[8], Stéphanie Clerc-Rosset[9], Graham William Knott[9], Jeffrey C Rathmell[10], E Dale Abel[11], Anders Meibom[4,12], Etienne Meylan[1,2]*

[1]Swiss Institute for Experimental Cancer Research (ISREC), School of Life Sciences, Ecole Polytechnique Fédérale de Lausanne, Lausanne, Switzerland; [2]Swiss Cancer Center Léman, Lausanne, Switzerland; [3]Bioinformatics Core Facility, Swiss Institute of Bioinformatics, Lausanne, Switzerland; [4]Laboratory for Biological Geochemistry, School of Architecture, Civil and Environmental Engineering, Ecole Polytechnique Fédérale de Lausanne, Lausanne, Switzerland; [5]Institute of Animal Pathology (COMPATH), University of Bern, CH-3012 Bern, and Histology Core Facility, School of Life Sciences, Ecole Polytechnique Fédérale de Lausanne, Lausanne, Switzerland; [6]Laboratory for Functional and Metabolic Imaging (LIFMET), Ecole Polytechnique Fédérale de Lausanne, Lausanne, Switzerland; [7]Center for Biomedical Imaging (CIBM), Ecole Polytechnique Fédérale de Lausanne, Lausanne, Switzerland; [8]Institute of Pathology, University of Bern, Bern, Switzerland; [9]BioEM Facility, School of Life Sciences, Ecole Polytechnique Fédérale de Lausanne, Lausanne, Switzerland; [10]Vanderbilt Center for Immunobiology, Vanderbilt University Medical Center, Nashville, United States; [11]Fraternal Order of Eagles Diabetes Research Center and Division of Endocrinology and Metabolism, Carver College of Medicine, University of Iowa, Iowa City, United States; [12]Center for Advanced Surface Analysis, Faculty of Geosciences and Environment, University of Lausanne, Lausanne, Switzerland

*For correspondence: etienne.meylan@epfl.ch

**Competing interests:** The authors declare that no competing interests exist.

**Abstract** Glucose utilization increases in tumors, a metabolic process that is observed clinically by $^{18}$F-fluorodeoxyglucose positron emission tomography ($^{18}$F-FDG-PET). However, is increased glucose uptake important for tumor cells, and which transporters are implicated in vivo? In a genetically-engineered mouse model of lung adenocarcinoma, we show that the deletion of only one highly expressed glucose transporter, Glut1 or Glut3, in cancer cells does not impair tumor growth, whereas their combined loss diminishes tumor development. $^{18}$F-FDG-PET analyses of tumors demonstrate that Glut1 and Glut3 loss decreases glucose uptake, which is mainly dependent on Glut1. Using $^{13}$C-glucose tracing with correlated nanoscale secondary ion mass spectrometry (NanoSIMS) and electron microscopy, we also report the presence of lamellar body-like organelles in tumor cells accumulating glucose-derived biomass, depending partially on Glut1. Our results demonstrate the requirement for two glucose transporters in lung adenocarcinoma, the dual blockade of which could reach therapeutic responses not achieved by individual targeting.

## Introduction

Glucose transporters are the first and rate-limiting step for cellular glucose utilization, a process often exacerbated in tumor cells that enables their growth and proliferation (*Ancey et al., 2018*; *Lunt and Vander Heiden, 2011*). Although inhibiting glucose metabolism in lung tumors could become an efficient treatment strategy (*Hensley et al., 2016*; *Patra et al., 2013*; *Xie et al., 2014*), whether and which glucose transporter(s) should be targeted remains unclear because of their possible functional redundancy. Additionally, in solid cancers, tumor cell growth can be fueled by nutrients other than glucose, or by using transporter-independent metabolic processes including autophagy and macropinocytosis (*Commisso et al., 2013*; *Karsli-Uzunbas et al., 2014*; *Romero et al., 2017*; *Son et al., 2013*).

In this study, we exploited the $Kras^{LSL-G12D/WT}$; $Trp53^{Flox/Flox}$ (KP) mouse model of lung adenocarcinoma to explore the importance of glucose transporters, expressed by tumor cells, in disease development. By glucose tracing and ultrastructural analyses, we identified the presence of lamellar body-like organelles in tumor cells, which are the primary site of glucose-derived biomass accumulation, occurring partly in a Glut1-dependent manner. We show that the deletion of Glut1 or Glut3 is not sufficient to decrease tumor progression, which is only affected significantly upon combined Glut1 and Glut3 loss.

## Results and discussion

To interrogate the importance of particular glucose transporters in the most frequent subtype of lung cancer, lung adenocarcinoma (LUAD), we used The Cancer Genome Atlas specific for LUAD (TCGA-LUAD) to compare gene expression within the facilitated glucose transporter (GLUT, gene name *SLC2A*) family. *SLC2A1* (GLUT1) followed by *SLC2A3* (GLUT3) were the most expressed members, (*Figure 1a*) and high *SLC2A1* expression was correlated with poor overall survival (*Figure 1— figure supplement 1a*). From a human tissue microarray comprising 18 cases of stage IB-IIB LUAD and 18 cases of stage IA-IIB lung squamous cell carcinoma (LUSC), GLUT1 protein was detected in all tumors. Specifically, it showed intermediate and strong GLUT1 expression in 44% (8/18) and 56% (10/18) of the LUAD lesions, respectively (*Figure 1b*). In LUSCs, GLUT1 exhibited strong staining in all samples (*Figure 1b*), confirming previous results obtained from squamous cell carcinomas of lung and other tissues (*Goodwin et al., 2017*; *Hsieh et al., 2019*). To obtain more information about *SLC2A1* expression in LUAD, we applied a non-negative matrix factorization of TCGA-LUAD samples, which generated four distinct clusters (NMF1-4, *Figure 1—figure supplement 1b*). *SLC2A1* and *SLC2A3* were the most expressed in NMF4, which contains the highest percentage of the *TP53* mutation (*Figure 1c*). In the immunocompetent $Kras^{LSL-G12D/WT}$; $Trp53^{Flox/Flox}$ (KP) mouse model where tumors consist of pure LUADs all harboring *Trp53* gene deletion, RNA sequencing analyses of bulk tumor samples revealed a predominance of NMF4-like lesions (*Figure 1—figure supplement 1c*). Glut1 protein levels in tumor cells gradually increased along tumor progression, with the majority of advanced grade lesions being positive with intermediate or strong Glut1 staining (*Figure 1— figure supplement 2a*). In normal lung, Glut1 was only weakly expressed or undetectable in the alveolar compartment that comprises alveolar type-1 and 2 (AT1, AT2) cells, endothelial cells and alveolar macrophages (*Figure 1—figure supplement 2b*), AT2 cells being considered as the principal tumor cell-of-origin in this model (*Desai et al., 2014*; *Sutherland et al., 2014*; *Xu et al., 2012*). In contrast, Glut1 was strongly expressed in the bronchiolar epithelial compartment that contains Club cells, another cell type permissive to tumor initiation (*Sutherland et al., 2014*; *Figure 1—figure supplement 2b*). To determine the contribution of Glut1 for tumor development, we crossed KP mice to $Slc2a1^{Flox/Flox}$ (G1) animals (*Young et al., 2011*), and initiated tumors in the resulting KPG1 mice and control KP mice by intratracheal lentiviral-Cre instillation (Lenti.PGK-Cre, *Figure 1—figure supplement 2c–d*). During tumor progression, we detected slightly but not significantly reduced changes in tumor growth rates monitored by X-rays micro-computed tomography (µCT) (*Figure 1d*). At sacrifice, although no significant difference was observed in the weight or in the number of lesions (*Figure 1e–f*), a significant reduction of high grade tumors (grades 4 and 5) was determined upon Glut1 loss in KP lesions (*Figure 1g*). Because there was no significant increase in KPG1 mouse survival (*Figure 1h*), these data together fail to reveal a clear impact of Glut1 deletion in the tumor epithelial cells on disease progression.

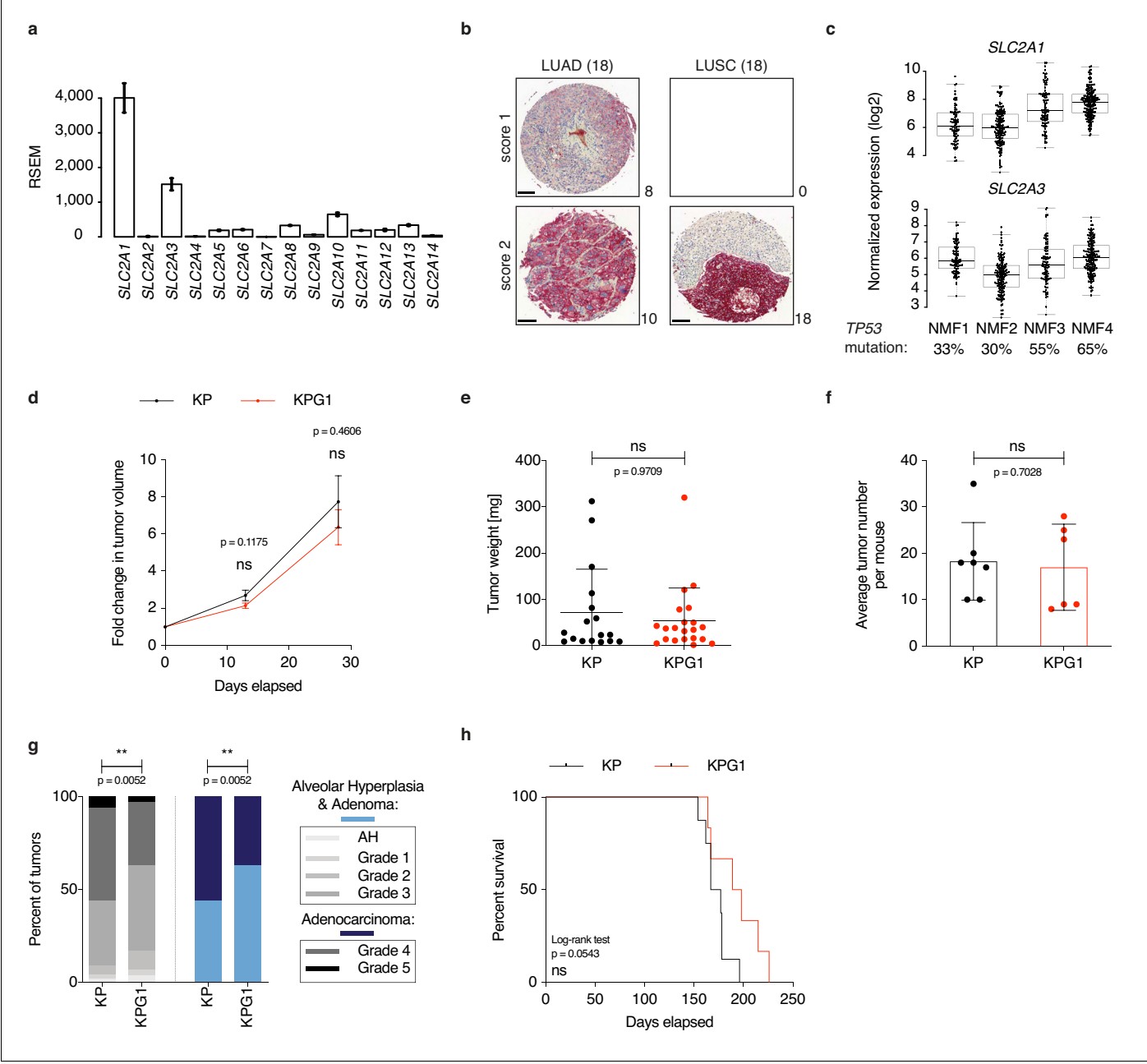

**Figure 1.** Glut1 deletion does not impact disease progression in *Kras*$^{LSL-G12D/WT}$; *Trp53*$^{Flox/Flox}$ mice. (a) Gene expression level (RSEM) of glucose transporters in TCGA-LUAD samples (n = 511). (b) Immunohistochemistry (IHC) from a next-generation tissue microarray of human LUAD and LUSC showing score 1 (intermediate) or score 2 (strong) GLUT1 staining. The number of cases per score and histology are indicated. Scale bars: 100 μm. (c) Expression level (log2 normalized) of *SLC2A1* and *SLC2A3* in the 4 NMF LUAD subtypes. Percent of *TP53* mutation in each subtype is indicated. (d) Graph with mean ± s.e.m. shows the fold changes of KP and KPG1 tumor volumes (n = 32 and 26, respectively) monitored during 28 days by μCT, starting at 16 weeks and 6 days post-tumor initiation with tumor volumes set to 1. ns: not significant by Mann-Whitney test. (e) Dot plot with mean ± s.d. shows KP and KPG1 tumor weights (n = 17 and 21) at sacrifice 29 weeks post-tumor initiation. ns: not significant by Mann-Whitney test. (f) Dot plot with mean ± s.d. shows the average number of KP and KPG1 tumors per mouse (n = 7 and 6 mice). ns: not significant by Mann-Whitney test. (g) Percent of KP (n = 128) and KPG1 (n = 102) lesions classified by tumor grades, either detailed from alveolar hyperplasia (AH) to grade 5 or discriminated between alveolar hyperplasia and adenomas, and adenocarcinomas. Alveolar hyperplasia and adenomas include the AH and the tumor grades 1, 2, and 3. Adenocarcinomas contain the tumor grades 4 and 5. **: p < 0.01. Fisher test was applied when comparing AH, grade 1, grade 2, grade 3, grade 4, and grade 5. Chi-square for trend was applied when comparing alveolar hyperplasia and adenomas, and adenocarcinomas. (h) Kaplan-Meier survival analysis of KP (n = 8) and KPG1 (n = 6) mice. ns: not significant by Log-rank test.

The online version of this article includes the following source data and figure supplement(s) for figure 1:

*Figure 1 continued on next page*

*Figure 1 continued*

**Source data 1.** Source files for tumor growth, grades and survival of KP and KPG1 mice.
**Figure supplement 1.** Non-negative matrix factorization generates four LUAD subtypes, with NMF4 being the closest to KP tumors.
**Figure supplement 2.** Glut1 is detectable in tumors and the bronchiolar epithelium but not in the alveolar compartment.
**Figure supplement 2—source data 1.** Source files for Glut1 protein expression analysis by KP tumor grade.

To detect if there are measurable changes in glucose utilization secondary to Glut1 deletion in vivo, we decided to monitor glucose-derived biomass accumulation in tumors with ultrastructural resolution. Specifically, we performed $^{13}$C-glucose injections followed by nanoscale secondary ion mass spectrometry (NanoSIMS) imaging (*Hoppe et al., 2013*) correlated with electron microscopy (EM) (*Figure 2a*). Intense $^{13}$C enrichments inside tumors revealed a compartmentalized intracellular accumulation of glucose-derived biomass, which can be attributed to a specific organelle of tumor cells visualized by the superposition of images obtained from scanning EM (SEM) and NanoSIMS (*Figure 2b*). From SEM and transmission EM (TEM), these organelles resemble lamellar bodies (LBs) based on their size, secretory behavior and morphology, which exhibit more or less packed, occasionally visible lamellae (*Figure 2b–c*). LBs are lipid-rich secretory organelles specifically produced in

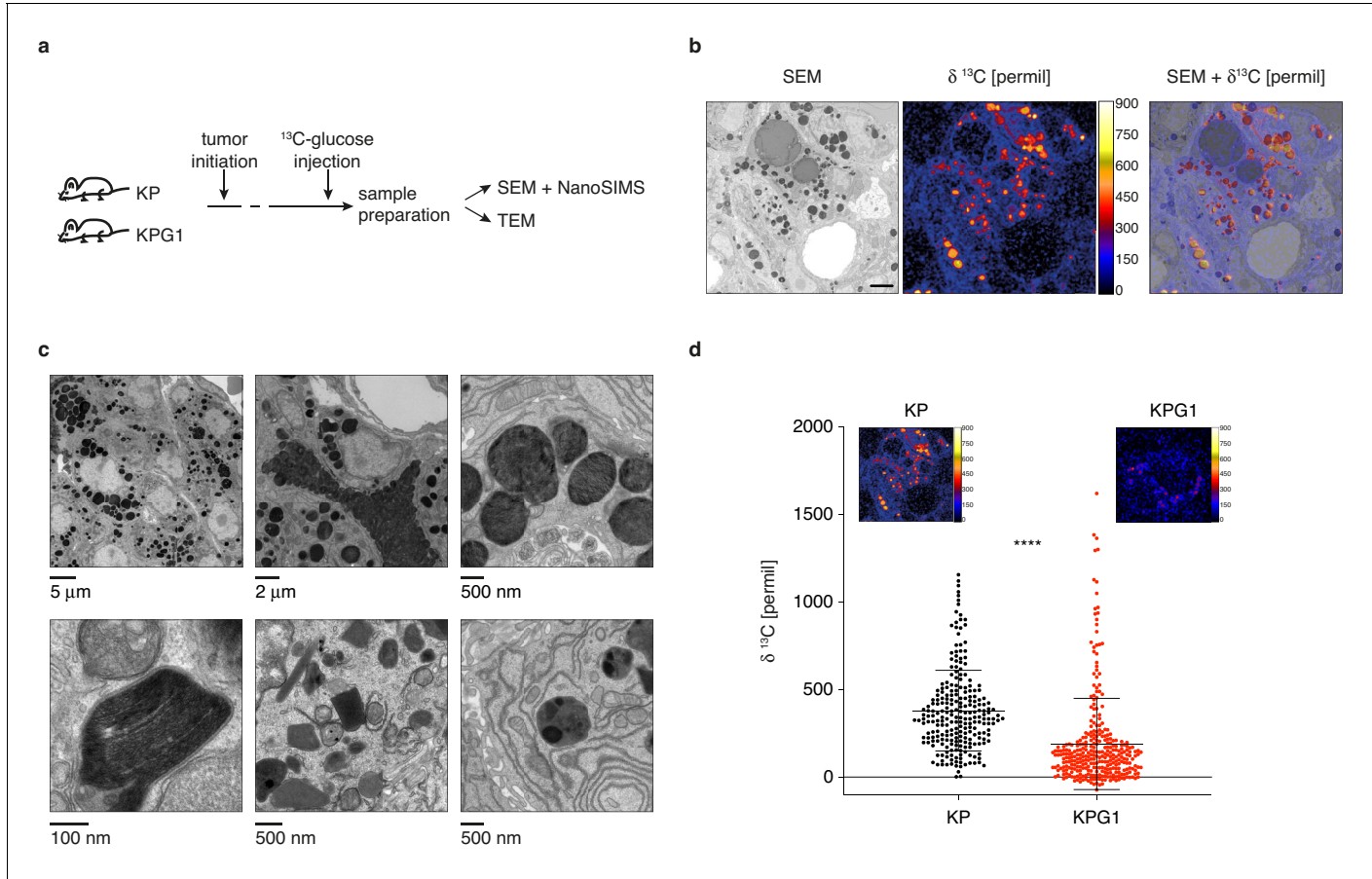

**Figure 2.** Lamellar body-like organelles are produced by tumor cells and accumulate most glucose-derived biomass. (a) Experimental setting. (b) Representative examples of a scanning electron microscopy (SEM) images (left), NanoSIMS isotopic image (middle) and their superposition (right) obtained from a KP tumor. Scale bar: 2 µm. (c) (upper panels) Representative transmission electron microscopy (TEM) images from KP tumors showing lamellar body-like organelles (LBOs) with different sizes, secretory behavior and presence of lamellae. (lower panels) High-resolution TEM micrographs showing high magnification lamellae (left), and LBOs with morphologies reminiscent of immature lamellar bodies (LBs) (middle and right). (d) Dot plot with mean ± s.d. showing the NanoSIMS quantification of $^{13}$C-enrichments in LBOs from KP and KPG1 tumors. Each dot represents the measurement of an individual LBO (n = 239 KP LBOs and n = 325 KPG1 LBOs). ****: p < 0.0001 by Mann-Whitney test. Representative NanoSIMS isotopic images of a (left) KP and (right) KPG1 tumor cell.

the lung by AT2 cells, responsible for surfactant production and release into the alveolar space. Within tumor cells, these LB-like organelles (LBOs) were typically from 200 nm to 2 μm in diameter, irregularly shaped and darkly stained. Often these contained more heavily stained regions and discernible tightly packed lamellae at higher magnifications (*Figure 2c*). Quantitative NanoSIMS images of these organelles revealed a lower $^{13}$C-enrichment signal intensity in KPG1 compared to KP tumor cells (*Figure 2d*), demonstrating the consequence of Glut1 loss in vivo.

Despite the differences measured by NanoSIMS, Glut1 deficiency only minimally affected tumor progression (see *Figure 1d–h*). We therefore hypothesized the existence of an alternative mechanism that sustains lung tumor growth. First, we isolated KP and KPG1 tumors, prepared single cell suspensions and tested their glycolytic capacity. In this *ex vivo* experimental setting, KPG1 tumors failed to stimulate glycolysis in response to glucose (*Figure 3a* and *Figure 3—figure supplement 1a*). As exception, one KPG1 tumor displayed the strongest glycolytic response of all tested tumors, which coincided with elevated expression of another high-affinity glucose transporter, Glut3 (*Figure 3a*). To understand better the mechanisms of Glut1-deficient tumor progression, we also performed RNA sequencing analyses from the non-immune (CD45$^-$) fraction of KP and KPG1 tumors. To increase the relevance of this comparison, we analysed three cohorts of mice, where tumors were initiated with Lenti.PGK-Cre, Ad5.SPC-Cre or Ad5.CC10-Cre, the latter two providing SPC$^+$- or CC10$^+$-cell restricted Cre expression for tumor initiation in distinct cell types (*Sutherland et al., 2014*) (see *Figure 1—figure supplement 2c*). Gene Set Enrichment Analysis (GSEA) highlighted, from all the Hallmark and KEGG up- and downregulated pathways, a unique significantly shared pathway between the three cohorts, which was upregulated in KPG1: 'Hallmark_bile acid metabolism' (*Figure 3—figure supplement 2a*). Upon closer examination of the genes included in this pathway, we identified several target genes of peroxisome proliferator-activated receptor alpha (PPARα) (*Figure 3b*), a crucial transcription factor in fatty acid catabolism (*Reddy and Hashimoto, 2001*). Furthermore, when interrogating the expression of known PPARα targets (*Rakhshandehroo et al., 2010*), GSEA highlighted their enrichment in KPG1 compared to KP tumors (*Figure 3c*). Thus, KPG1 tumors have a stronger expression of known PPARα-target genes than KP lesions. Together, our metabolic and molecular analyses suggested two possible and distinct routes for Glut1-deficient tumor growth: the uptake of the same nutrient through another glucose transporter, Glut3, or *via* a PPARα-dependent metabolic shift (*Figure 3d*).

To test the hypothesis of an involvement of PPARα in lung tumor progression, we cloned a mouse PPARα-dominant negative construct, PPARαΔ13 (*Michalik et al., 2005*), into a bi-promoter Lenti. Cre vector allowing doxycycline-inducible expression (Lenti.*TRE*-PPARαΔ13_PGK-Cre) in KP and KPG1 tumors (mice were crossed to *CCSP-rtTA* transgenic mice [*Meylan et al., 2009*; *Figure 3—figure supplement 3a–b*]). Doxycycline-mediated PPARαΔ13 induction in KP mice led only to a trend toward a reduction in tumor growth rates, comparable to the growth of KPG1 versus KP tumors. Within the KPG1 group, PPARαΔ13 induction did not further reduce tumor growth (*Figure 3—figure supplement 3c*). Global gene expression analyses from tumor samples confirmed the functionality of the PPARαΔ13 construct in both KP and KPG1 tumors, as 'Hallmark_bile acid metabolism', 'Hallmark_fatty acid metabolism', 'KEGG_peroxisome' and 'KEGG_ fatty acid metabolism' pathways were repressed in tumors harboring this construct compared to tumors from KP-rtTA or KPG1-rtTA mice not fed on a doxycycline diet (*Figure 3—figure supplement 3d*). Thus, although PPARαΔ13 downregulates PPARα-related pathways in tumors, it fails to affect their growth even in absence of Glut1.

To test our second hypothesis, we initially monitored Glut3 expression in tumors. By real-time PCR, *Slc2a3* was expressed to varying levels but not significantly differently between KP and KPG1 tumors (*Figure 4—figure supplement 1a*). In contrast, *Slc2a3* was almost undetectable in tumors from *Kras$^{LSL-G12D}$; Stk11$^{Flox/Flox}$* (KL) mice (*Figure 4—figure supplement 1a*), another lung cancer model where Lkb1 (*Stk11*)-deficient tumors progressing from an adeno- to a squamous histology become exquisitely Glut1-dependent (*Hsieh et al., 2019*). In absence of Glut1 (KLG1), *Slc2a3* mRNA expression remained very low (*Figure 4—figure supplement 1a*). Accordingly, Glut3 protein was undetectable by immunohistochemistry in KL and KLG1 tumors, whereas it was expressed in KP and KPG1 lesions (*Figure 4—figure supplement 1b*). More specifically, Glut3 stained at the tumor cell membrane in a minority (37%) of small and a majority (76%) of big KP lesions (*Figure 4—figure supplement 1c*). Furthermore, within Glut3-expressing KP tumors, areas positive for Glut3 frequently correlated with a stronger expression of Glut1 (*Figure 4a* and *Figure 4—figure supplement 1d*),

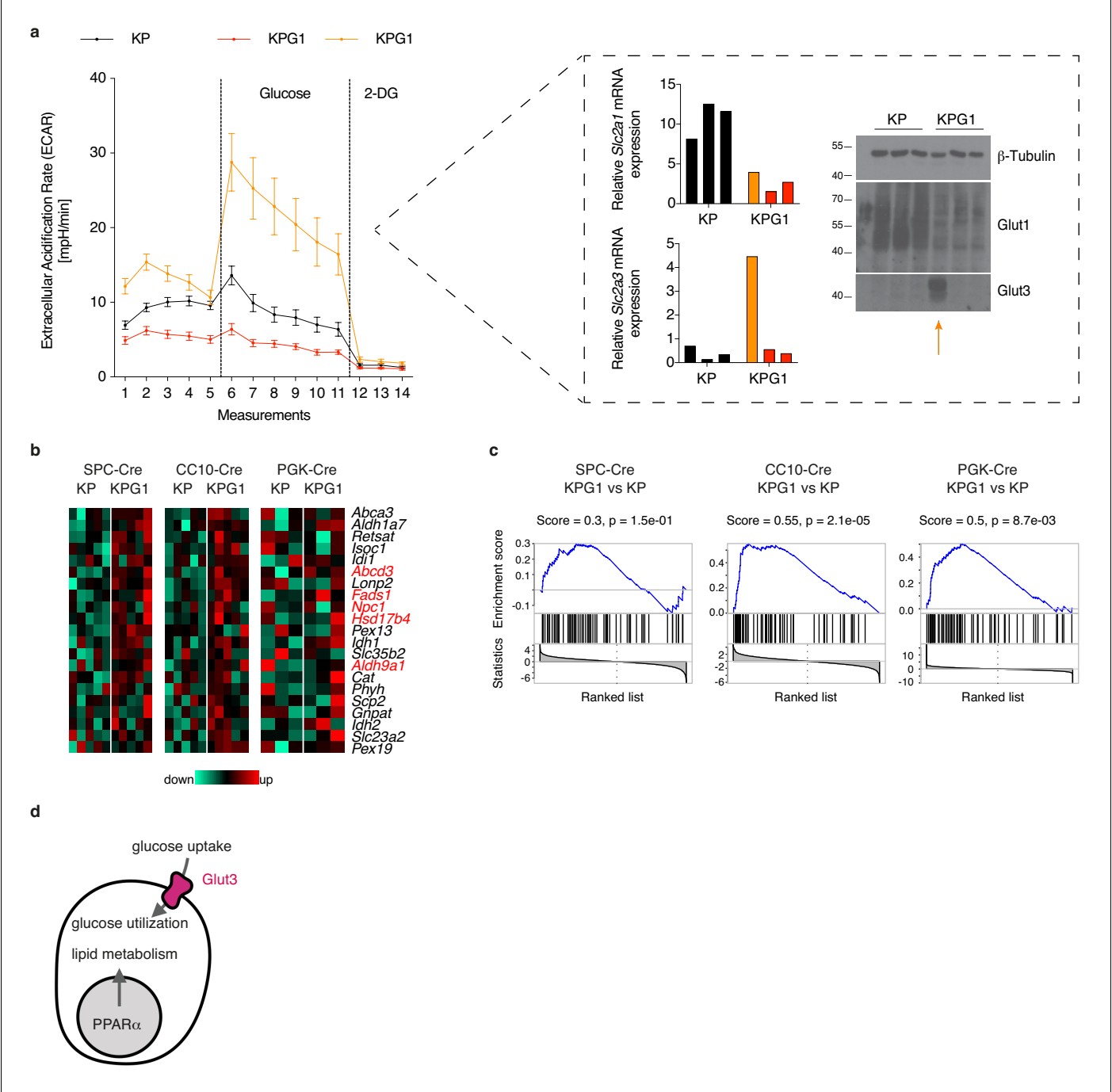

**Figure 3.** Glut1-deficient tumor analyses suggest Glut3- or PPARα-dependent compensatory mechanisms. (**a**) (left) Tumor-derived single cell preparations were placed in a Seahorse XF analyzer and subjected to longitudinal extracellular acidification rate (ECAR) measurements. Glucose or 2-deoxyglucose (2-DG) were added where indicated. Data are means ± s.e.m. of 3 KP and 3 KPG1 tumor-derived single cells each analyzed in ten or five technical replicates. The data from the outlier KPG1 are shown separately (orange) from the 2 other KPG1 tumors (red). (right) Real-time PCR and western blot analyses of Glut1 and Glut3 from the samples analyzed by Seahorse. (**b**) Bile acid metabolism genes from Hallmark collection (mSigDB) induced in KPG1 compared to KP from SPC-Cre, CC10-Cre and PGK-Cre tumors. Expression above the median is shown in red, below in green. Genes in red are PPARα targets (*Rakhshandehroo et al., 2010*). (**c**) Gene Set Enrichment Analysis (GSEA) of PPARα targets on genes ranked based on t-statistics when comparing KPG1 and KP tumors in SPC-Cre, CC10-Cre and PGK-Cre. p-values were obtained by normalized enrichment score (NES) of $10^5$ random permutations. (**d**) Model illustrating two possible alternatives to Glut1 deficiency for tumor growth.

The online version of this article includes the following source data and figure supplement(s) for figure 3:

**Source data 1.** Source files for Seahorse analysis of KP and KPG1 tumors.

*Figure 3 continued*

**Figure supplement 1.** Glut1 deletion compromises *ex vivo* glycolytic metabolism.
**Figure supplement 1—source data 1.** Source files for Seahorse analysis of KP and KPG1 tumors.
**Figure supplement 2.** Hallmark 'bile acid metabolism' is upregulated in Glut1-deficient tumors.
**Figure supplement 3.** PPARαΔ13 does not interfere with tumor development of KP and KPG1 mice.
**Figure supplement 3—source data 1.** Source files for tumor growth of KP and KPG1 mice, with or without PPARαΔ13 induction.

suggesting intra-tumor heterogeneity in glucose utilization in this mouse model, as demonstrated earlier in human non-small cell lung cancer (*Hensley et al., 2016*). We then crossed KP mice to *Slc2a3^Flox/Flox* mice (generating KPG3 mice, see *Figure 1—figure supplement 2c–d*). When comparing tumor development in KP and KPG3 animals, we failed to detect differences in overall tumor growth rates, *ex vivo* glycolysis, mouse survival, or tumor grade distribution (*Figure 4—figure supplement 2a–d*). Although this indicates that Glut3 deletion alone does not affect tumors in a detectable manner, we next wanted to evaluate its importance in a Glut1-deficient context. To do this, we generated KPG1G3 mice by interbreeding KP mice with *Slc2a1^Flox/Flox* and *Slc2a3^Flox/Flox* mice (see *Figure 1—figure supplement 2c–d*). Deletion of the two glucose transporters, Glut1 and Glut3, from the tumor epithelial cell compartment resulted in a decreased number of tumors (*Figure 4b*). Furthermore, contrasting with our data obtained from each separately-targeted Gluts, combined Glut1 and Glut3 deletion led to a significant reduction of tumor growth rates monitored by µCT scans, to a reduced tumor size with an absence of big (>50 mg) lesions at sacrifice, to a decrease of high grade tumors, and to an extended mouse survival (*Figure 4c–g* and *Figure 4—figure supplement 3a–b*). This synergistic action of the dual Glut1 and Glut3 deletion was also demonstrated in vitro with four different human NSCLC cell lines, on which the strongest reduction of tumor cell viability resulted from the combined knockdown of these glucose transporters (*Figure 4—figure supplement 3c–f*). Together, these data position Glut1 and Glut3 as a coordinated glucose transport unit required to fuel the growth of LUAD.

To trace and compare directly glucose uptake in tumors from the different mouse genotypes, we used positron emission tomography with $^{18}$F-fluorodeoxyglucose ($^{18}$F-FDG-PET) in KP, KPG1, KPG3, and KPG1G3 tumors. The maximum standardized uptake value (SUV$_{max}$) of each tumor revealed that Glut1-deficient KP tumors incorporated significantly less $^{18}$F-FDG than control lesions (*Figure 4h–i*), which strengthens the NanoSIMS data (see *Figure 2d*). Glut3-deficient and control KP tumors presented a similar $^{18}$F-FDG absorption (*Figure 4h–i*). This result might be explained by the fact that Glut3 positive KP tumors mainly co-express Glut1 and Glut3 (see *Figure 4—figure supplement 1d*). Thus, in absence of Glut3, lesions might certainly incorporate $^{18}$F-FDG *via* Glut1. The $^{18}$F-FDG absorption of Glut1- and Glut3-deficient KP tumors was significantly reduced as compared to control KP lesions (*Figure 4h–i*). Because glucose uptake was not significantly lower in KPG3 compared to KP tumors, and in KPG1G3 compared to KPG1 tumors, a possible explanation is that most tumors analyzed had not yet gained Glut3 expression and become dependent on it, which agrees with an elevated Glut3 expression specifically in big lesions (see *Figure 4—figure supplement 1c*). Finally, to investigate if Glut1- and Glut3-deficient tumors might incorporate glucose *via* another glucose transporter, we performed a gene expression analysis of glucose transporter family members on KP and KPG1G3 lesions. This analysis showed that the expression of no other glucose transporter was changed upon dual Glut1 and Glut3 deletion in KP tumors (*Figure 4—figure supplement 4a*). Altogether, these data demonstrate that deleting Glut1 and Glut3 in KP tumors reduces glucose uptake, and that Glut1 highly contributes to glucose uptake in KP tumors. These results may also lead to the hypothesis that at least a part of the remaining $^{18}$F-FDG-PET signal detected in KPG1G3 tumors might come from glucose uptake by the cells of the tumor immune microenvironment.

Recently, pharmacological inhibition of a secondary active, Na$^+$-dependent glucose transporter, Sglt2, showed its role in premalignant KP lung lesions (*Scafoglio et al., 2018*), while a glycolytic switch was identified in advanced compared to early tumors (*Kerr et al., 2016*). These studies, together with our gene targeting approaches, definitively demonstrate the importance of glucose utilization for LUAD development in vivo. In the KL model, Glut1 deletion impairs LUSC development (*Hsieh et al., 2019*), and we find that Lkb1-deficient lung tumors lack detectable Glut3, precluding any compensation occurring through this transporter. In contrast, in pure LUAD our results

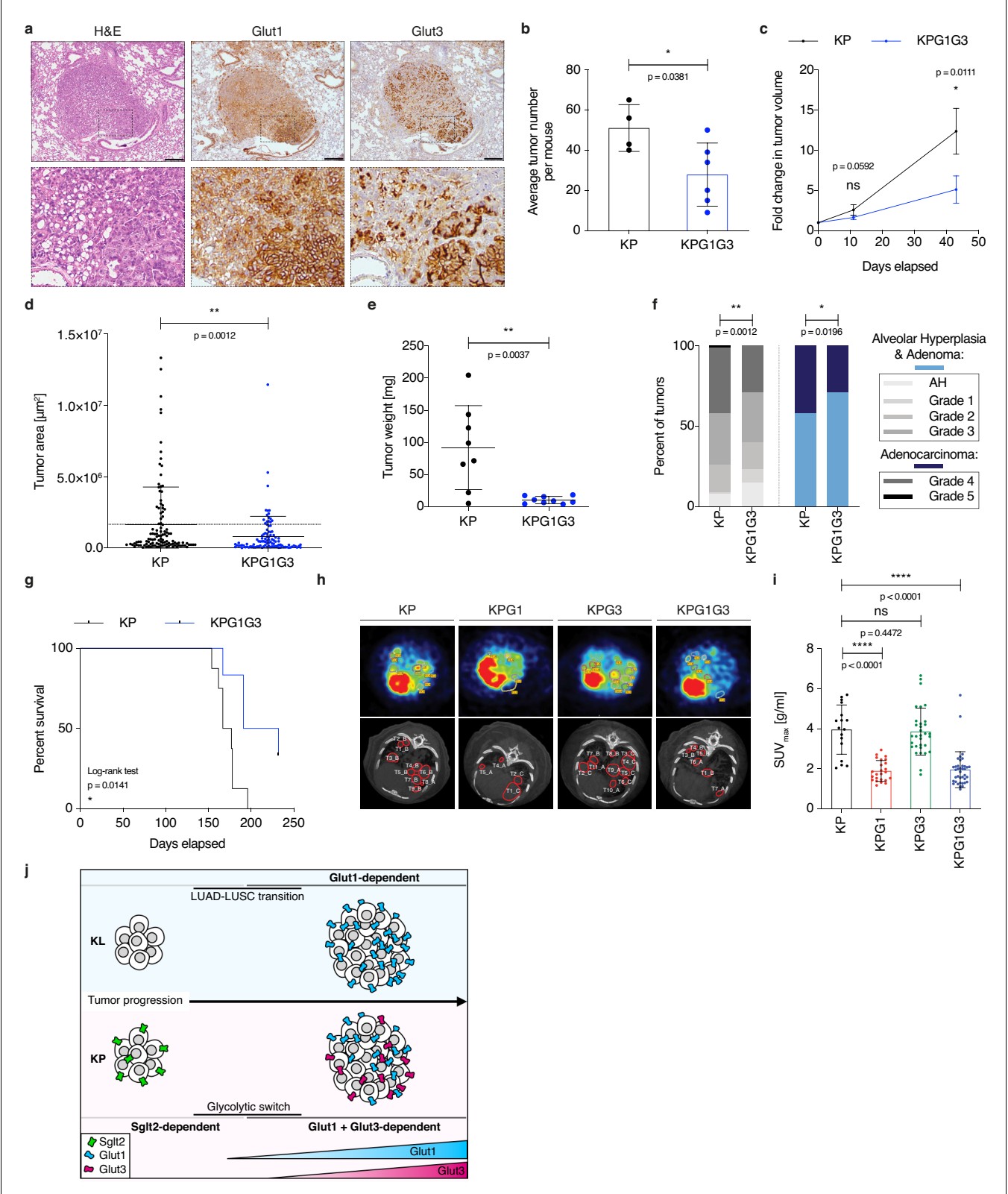

**Figure 4.** Combined Glut1 and Glut3 deletion impairs lung tumor growth. (**a**) Representative examples of H&E, Glut1 or Glut3 staining by IHC from serial sections of a KP tumor showing Glut1-3 co-localization. Scale bars: 200 μm. (**b**) Histogram with mean ± s.d. shows the average number of KP and KPG1G3 tumors per mouse (n = 4 and 6 mice). *: p < 0.05 by Mann-Whitney test. (**c**) Graph with mean ± s.e.m. shows the fold changes of KP and KPG1G3 tumor volumes (n = 8 and 9 tumors) monitored during 43 days by μCT, starting at 15 weeks and 2 days post-tumor initiation with tumor

*Figure 4 continued on next page*

*Figure 4 continued*

volumes set to 1. ns: not significant by Mann-Whitney test; *: p < 0.05 by Mann-Whitney test. (d) Dot plot with mean ± s.d. shows tumor areas in KP and KPG1G3 mice (n = 115 and 101 tumors) calculated from lung sections. **: p < 0.01 by Mann-Whitney test. (e) Dot plot with mean ± s.d. shows KP and KPG1G3 tumor weights (n = 8 and 9 tumors, respectively) at sacrifice. **: p < 0.01 by Mann-Whitney test. (f) Percent of KP (n = 150) and KPG1G3 (n = 170) lesions classified by tumor grades, either detailed from alveolar hyperplasia (AH) to grade 5 or discriminated between alveolar hyperplasia and adenomas, and adenocarcinomas. Alveolar hyperplasia and adenomas include the AH and the tumor grades 1, 2, and 3. Adenocarcinomas contain the tumor grades 4 and 5. *: p < 0.05; **: p < 0.01. Fisher test was applied when comparing AH, grade 1, grade 2, grade 3, grade 4, and grade 5. Chi-square for trend was applied when comparing alveolar hyperplasia and adenomas, and adenocarcinomas. (g) Kaplan-Meier survival analysis of KP (n = 8) and KPG1G3 (n = 6) mice. *: p < 0.05 by Log-rank test. (h) Representative (upper panels) positron emission tomography (PET) scan and (lower panels) μCT scan images illustrating the $^{18}$F-FDG absorption of KP, KPG1, KPG3 and KPG1G3 tumors. (i) Dot plot with mean ± s.d. displays maximum standardized uptake value (SUV max ) of KP (n = 17), KPG1 (n = 24), KPG3 (n = 32) and KPG1G3 (n = 38) lesions. ns: not significant by Mann-Whitney test; ****: p < 0.0001 by Mann-Whitney test. (j) Model illustrating the dependency on glucose transporters for KP and KL tumor progression.

The online version of this article includes the following source data and figure supplement(s) for figure 4:

**Source data 1.** Source files for tumor growth, grades and survival of KP and KPG1G3 mice.
**Figure supplement 1.** Glut1 and Glut3 co-localize frequently within tumors.
**Figure supplement 2.** Glut3 deletion alone does not affect lung tumor development.
**Figure supplement 2—source data 1.** Source files for tumor growth, Seahorse analysis, survival and grades of KP and KPG3.
**Figure supplement 3.** Dual knockdown of GLUT1 and GLUT3 most strongly reduces human non-small cell lung cancer cell viability *in vitro*, corroborating the *in vivo* data.
**Figure supplement 3—source data 1.** Source files for tumor growth of KP and KPG1G3 mice.
**Figure supplement 4.** KP tumors do not express differently another glucose transporter upon dual Glut1 and Glut3 deletion.

indicate that *Trp53*-deficient tumor cells depend on glucose uptake based on a redundant system, whereby the expression of either Glut1 or Glut3 is sufficient to sustain tumor growth (*Figure 4j*). For clinical perspectives, we anticipate that dual Glut1 and Glut3 blockade will be necessary to achieve strong anti-tumor efficacy.

# Materials and methods

## Key resources table

| Reagent type (species) or resource | Designation | Source or reference | Identifiers | Additional information |
|---|---|---|---|---|
| Strain, strain background (*M. musculus*, male, female) | K | The Jackson Laboratory | RRID:IMSR_JAX:008179 | *Kras*$^{LSL-G12D/WT}$ *Trp53*$^{Flox/Flox}$ K and P interbred at EPFL |
| Strain, strain background (*M. musculus*, male, female) | P | The Jackson Laboratory | RRID:IMSR_JAX:008462 | *Kras*$^{LSL-G12D/WT}$ *Trp53*$^{Flox/Flox}$ K and P interbred at EPFL |
| Strain, strain background (*M. musculus*, male, female) | KPG1 | Interbred at EPFL | | *Kras*$^{LSL-G12D/WT}$ *Trp53*$^{Flox/Flox}$ *Slc2a1*$^{Flox/Flox}$ |
| Strain, strain background (*M. musculus*, male, female) | KPG3 | Interbred at EPFL | | *Kras*$^{LSL-G12D/WT}$ *Trp53*$^{Flox/Flox}$ *Slc2a3*$^{Flox/Flox}$ |
| Strain, strain background (*M. musculus*, male, female) | KPG1G3 | Interbred at EPFL | | *Kras*$^{LSL-G12D/WT}$ *Trp53*$^{Flox/Flox}$ *Slc2a1*$^{Flox/Flox}$ *Slc2a3*$^{Flox/Flox}$ |
| Strain, strain background (*M. musculus*, male, female) | KPR | Interbred at EPFL | | *Kras*$^{LSL-G12D/WT}$ *Trp53*$^{Flox/Flox}$ *CCSP-rtTA* |

*Continued on next page*

*Continued*

| Reagent type (species) or resource | Designation | Source or reference | Identifiers | Additional information |
|---|---|---|---|---|
| Strain, strain background (*M. musculus*, male, female) | KPRG1 | Interbred at EPFL | | $Kras^{LSL-G12D/WT}$ $Trp53^{Flox/Flox}$ $Slc2a1^{Flox/Flox}$ CCSP-rtTA |
| Strain, strain background (*M. musculus*, male, female) | KL | Interbred at EPFL | | $Kras^{LSL-G12D/WT}$ $Stk11^{Flox/Flox}$ |
| Strain, strain background (*M. musculus*, male, female) | KLG1 | Interbred at EPFL | | $Kras^{LSL-G12D/WT}$ $Stk11^{Flox/Flox}$ $Slc2a1^{Flox/Flox}$ |
| Cell line (*Homo sapiens*) | A549 | ATCC | RRID:CVCL_0023 | Authentication: 100% |
| Cell line (*Homo sapiens*) | Calu-6 | ATCC | RRID:CVCL_0236 | Authentication: 100% |
| Cell line (*Homo sapiens*) | H460 | ATCC | RRID:CVCL_0459 | Authentication: 100% |
| Cell line (*Homo sapiens*) | SW 1573 | ATCC | RRID:CVCL_1720 | Authentication: 93.3% |
| Transfected construct (*Homo sapiens* and *M. musculus*) | siCTR | Invitrogen | Cat# 4390843 | Transfected construct |
| Transfected construct (*Homo sapiens*) | si*SLC2A1*#1 | ThermoFisher Scientific | s12926 Cat# 4390824 | Transfected construct |
| Transfected construct (*Homo sapiens*) | si*SLC2A1*#2 | ThermoFisher Scientific | s12927 Cat# 4390824 | Transfected construct |
| Transfected construct (*Homo sapiens*) | si*SLC2A3*#1 | ThermoFisher Scientific | s12931 Cat# 4390824 | Transfected construct |
| Transfected construct (*Homo sapiens*) | si*SLC2A3*#2 | ThermoFisher Scientific | s12932 Cat# 4390824 | Transfected construct |
| Antibody | GLUT1 (Rabbit monoclonal) | Abcam | RRID:AB_10903230 | IHC (1:650) |
| Antibody | GLUT1 (Rabbit polyclonal) | Millipore | RRID:AB_1587074 | IHC (ngTMA) (1:5000); WB (1:1000) |
| Antibody | GLUT3 (Rabbit polyclonal) | Abcam | RRID:AB_732609 | IHC (1:200); WB (1:1000) |
| Antibody | PPARα (Rabbit polyclonal) | ThermoFisher Scientific | RRID:AB_2165595 | WB (1:1000) |
| Antibody | β-tubulin (Rabbit polyclonal) | Santa Cruz Biotechnology | RRID:AB_2241191 | WB (1:1000) |
| Antibody | γ-tubulin (Mouse monoclonal) | Sigma-Aldrich | RRID:AB_477584 | WB (1:10000) |
| Recombinant DNA reagent | rtTA (plasmid) | Produced at EPFL | | |
| Recombinant DNA reagent | Lenti.PGK-Cre (Viral construct) | Produced at EPFL | | |
| Recombinant DNA reagent | Lenti.*TRE*-PPARαΔ13_PGK-Cre (Viral construct) | Produced at EPFL | | Cloned from PPARαΔ13 plasmid (L. Michalik, UNIL, Lausanne) |

*Continued on next page*

*Continued*

| Reagent type (species) or resource | Designation | Source or reference | Identifiers | Additional information |
|---|---|---|---|---|
| Recombinant DNA reagent | Ade5.SPC-Cre (Viral construct) | University of Iowa Viral Vector Core Facility | Ade5. SPC-Cre | |
| Recombinant DNA reagent | Ade5.CC10-Cre (Viral construct) | University of Iowa Viral Vector Core Facility | Ade5. CC10-Cre | |
| Sequence-based reagent | *Rpl30 (M. musculus)* | ThermoFisher Scientific | Mm01611464_g1 Cat# 4331182 | Taqman probe (Housekeeping gene) |
| Sequence-based reagent | *Slc2a1 (M. musculus)* | ThermoFisher Scientific | Mm00441480_m1 Cat# 4331182 | Taqman probe |
| Sequence-based reagent | *Slc2a2 (M. musculus)* | ThermoFisher Scientific | Mm00446229_m1 Cat# 4331182 | Taqman probe |
| Sequence-based reagent | *Slc2a3 (M. musculus)* | ThermoFisher Scientific | Mm00441483_m1 Cat# 4331182 | Taqman probe |
| Sequence-based reagent | *Slc2a4 (M. musculus)* | ThermoFisher Scientific | Mm00436615_m1 Cat# 4331182 | Taqman probe |
| Sequence-based reagent | *Slc2a5 (M. musculus)* | ThermoFisher Scientific | Mm00600311_m1 Cat# 4331182 | Taqman probe |
| Sequence-based reagent | *Slc2a6 (M. musculus)* | ThermoFisher Scientific | Mm00554217_m1 Cat# 4331182 | Taqman probe |
| Sequence-based reagent | *Slc2a7 (M. musculus)* | ThermoFisher Scientific | Mm01260610_m1 Cat# 4331182 | Taqman probe |
| Sequence-based reagent | *Slc2a8 (M. musculus)* | ThermoFisher Scientific | Mm00444634_m1 Cat# 4331182 | Taqman probe |
| Sequence-based reagent | *Slc2a9 (M. musculus)* | ThermoFisher Scientific | Mm00455122_m1 Cat# 4331182 | Taqman probe |
| Sequence-based reagent | *Slc2a10 (M. musculus)* | ThermoFisher Scientific | Mm01249519_m1 Cat# 4331182 | Taqman probe |
| Sequence-based reagent | *Slc2a12 (M. musculus)* | ThermoFisher Scientific | Mm02375931_s1 Cat# 4331182 | Taqman probe |
| Sequence-based reagent | *Slc2a13 (M. musculus)* | ThermoFisher Scientific | Mm01306489_m1 Cat# 4331182 | Taqman probe |
| Sequence-based reagent | *Slc5a2* (Sglt2) *(M. musculus)* | ThermoFisher Scientific | Mm00453831_m1 Cat# 4331182 | Taqman probe |
| Sequence-based reagent | *Slc50a1* (Sweet) *(M. musculus)* | ThermoFisher Scientific | Mm00485707_m1 Cat# 4331182 | Taqman probe |
| Commercial assay or kit | CD45 microbeads | Miltenyi Biotec | 130-052-301 | CD45 magnetic cell sorting |
| Chemical compound, drug | Doxycycline diet (dox) | Kliba Nafag | 3242 diet + DOX 0.625 g kg$^{-1}$ | 625 mg kg$^{-1}$ |
| Chemical compound, drug | Doxycycline (dox) | Sigma-Aldrich | D9891 | 2 µg ml$^{-1}$ |
| Software, algorithm | Analyze 12.0 | PerkinElmer | Analyze 12.0 | Tumor volume analysis |
| Software, algorithm | OsiriX MD | Pixmeo | RRID:SCR_013618 | Tumor volume analysis |
| Software, algorithm | QuPath | QuPath | RRID:SCR_018257 DOI: https://doi.org/10.1038/s41598-017-17204-5 | |

## TCGA data processing

The Cancer Genome Atlas (TCGA) Lung Adenocarcinoma (LUAD) dataset was retrieved from http://cancergenome.nih.gov. Samples with clinical information and mRNA (RNASEQV2) expression data

were selected (511 samples including 448 samples with survival data), and the data were downloaded using the RTCGAtoolbox (v2.12.1). RSEM gene quantifications as provided by TCGA were taken, counts were converted to log2 normalized counts expression and batch effect was removed using voom and removeBatchEffect functions from the limma package (v3.38.3).

## Non-negative matrix factorization

Non-negative matrix factorization (NMF) was done with the NMF package (v0.21.0) and standard strategies applied to the matrix in 511 normalized TCGA-LUAD samples. Negative values present in the dataset were zeroed to obtain a non-negative matrix, and the NMF was performed for ranks 2 to 6 using the Brunet algorithm. Two hundred runs were performed per rank with random seeding. For each rank, the cophenetic correlation coefficient and silhouette score were computed from the 200 solutions and the factorization achieving the minimum residual error was kept as the solution for further analyses. The cophenetic correlation coefficients and silhouette scores were directly obtained using the plot function of the NMF package. As a control, the NMF was also performed on a randomized dataset built by taking within each patient a random permutation of the gene expression values, using the randomize function of the NMF package. After having selected rank four as the best one, the NMF was rerun only for rank four and for a superior number of runs (500 runs). The final NMF-based matrix is composed of 935 metagenes and 511 samples split into four subtypes (shown as a heatmap). Correspondence to the three subtypes found by the Cancer Genome Atlas Research Network (*The Cancer Genome Atlas Research Network, 2014*) (proximal proliferative, proximal inflammatory and terminal respiratory unit) and presence of *TP53* mutation are shown on top bars.

To identify which human NMF subtype corresponds the most to KP mouse bulk Lenti tumors, we combined mouse and human normalized expression data, removed batch effect using ComBat (package sva v3.30.1), fitted a logistic regression to the human data set using glmnet package (v2.0–18) and 10-fold cross-validation with default parameters, and applied the fitted model as prediction method to the mouse data. Human and mouse orthologous genes were obtained using biomaRt (v2.38.0) and only genes in common between mouse and human were used.

## Human survival analysis

Univariate analyses of overall survival (OS) were performed using Cox proportional hazard regression models available in the survival package (v2.44–1.1). Survival curves were computed with the Kaplan-Meier method. p-values were computed with Wald test.

## Study approval

All mouse experiments were performed with the permission of the Veterinary Authority of Canton de Vaud, Switzerland (license numbers: VD2391 and VD2663).

## Animal models

*Kras*^LSL-G12D/WT^ (K) (RRID:IMSR_JAX:008179) and *Trp53*^Flox/Flox^ mice (P) (RRID:IMSR_JAX:008462) in a C57BL6/J background were purchased from The Jackson Laboratory, and were crossed to obtain *Kras*^LSL-G12D/WT^; *Trp53*^Flox/Flox^ (KP) mice. *Stk11*^Flox/Flox^ mice in a mixed background (FVB; 129S6) were obtained from R. DePinho (The University of Texas MD Anderson Cancer Center) through the National Cancer Institute mouse repository, were backcrossed seven times to C57BL6/J and were bred with *Kras*^LSL-G12D/WT^ mice to obtain *Kras*^LSL-G12D/WT^; *Stk11*^Flox/Flox^ (KL) mice. The generation of the *Slc2a1*^Flox/Flox^ (G1) mice and *Slc2a1* genotyping were described previously (*Young et al., 2011*). *Slc2a3*^Flox/Flox^ (G3) mice were produced at the Duke University Transgenic Facility upon blastocyst microinjection of targeted ES cells obtained from KOMP Repository (project CSD48048). The resulting mice were then crossed to Flp transgenic mice to remove the neomycin resistance cassette and generate mice with a *Slc2a3*^Flox^ allele, in which exon six is flanked by LoxP sites. *Slc2a3* genotyping was performed by PCR (94°C 5 min, followed by 40 cycles of 94°C 30 s, 56°C 30 s and 72°C 30 s, followed by 72°C 7 min) using oligos 5′-TAGTGCCCAGGAATGTGAGGTCAG-3′ and 5′-GCCCCAC-CAGATTTACCAAAGG-3′. While heterozygous mice were identified this way, in cases with only one detected band, DNA sequencing was further performed to discriminate without ambiguity between wild-type and floxed using 5′-TAGTGCCCAGGAATGTGAGGTCAG-3′.

KP mice were bred to G1 or G3 to obtain KPG1, KPG3 or KPG1G3 mice. KL mice were bred to G1 to obtain KLG1 mice. Control mice used throughout the study carry two wild-type (WT) copies of *Slc2a1* (or *Slc2a3*) or are heterozygotes (*Slc2a1*^*WT/Flox*^), as we did not notice any difference of tumor development comparing *Slc2a1*^*WT/WT*^ to *Slc2a1*^*WT/Flox*^ conditions. CCSP-rtTA mice were obtained from The Jackson Laboratory and were crossed to KP and KPG1 mice to enable doxycycline-mediated PPARαΔ13 transgene expression. For transgene induction, mice were fed on a diet of doxycycline-containing pellets (625 mg kg$^{-1}$; 3242 diet + DOX 0.625 g kg$^{-1}$; Kliba Nafag) for 15 days, or on a regular diet as control.

## Viral constructs

The dominant-negative mouse PPARα construct, PPARαΔ13 (*Michalik et al., 2005*), was cloned into bi-promoter Lenti.Cre vectors for doxycycline-inducible expression (Lenti.*TRE*-PPARαΔ13_PGK-Cre). Lentivirus preparation and titration have been described before (*Faget et al., 2017*). Lenti.*TRE*-PPARαΔ13_PGK-Cre vectors were intratracheally instilled at 6000 lentiviral units per mouse. Lenti.PGK-Cre vectors were used either at 3000 lentiviral units per mouse or at 4000 lentiviral units per mouse. Adenoviral vectors Ad5.SPC-Cre and Ad5.CC10-Cre were purchased from the University of Iowa Viral Vector Core Facility and used for intratracheal instillation at 2.10$^8$ PFU per mouse.

## Tumor volume measurements

Longitudinal tumor volume monitoring was performed by X-rays micro-computed tomography (μCT; Quantum FX; PerkinElmer). Mice were anaesthetized using isoflurane (Piramal, 56.761.002) before the scanning procedure. Animals were then maintained under anesthesia during lung imaging set at 50 μm voxel size, with retrospective gating. Individual tumor volumes were measured and calculated either with Analyze 12.0 software (PerkinElmer) or with OsiriX MD software (Pixmeo; RRID:SCR_013618).

## Mouse survival analysis

Overall mouse survival was evaluated considering actual mouse death or when lung tumor burden compelled to sacrifice. Control KP mice were the same for the three Kaplan-Meier analyses performed.

## Assessment of tumor parameters

Tumor grading was assessed from haematoxylin and eosin (H and E) stained lung sections by an experienced veterinary pathologist (CG), based on a previous classification system (*Jackson et al., 2005*). Briefly, lesions were categorized as alveolar hyperplasia (AH), or as tumor on a 1–5 severity grading scale from grade one adenomas to grade five adenocarcinomas. Glut1 staining (defined as weak, intermediate or strong) was monitored in all grades. Tumor numbers were counted from lung sections. Tumor areas were calculated from lung sections using QuPath (*Bankhead et al., 2017*; RRID:SCR_018257). To evaluate Glut3 staining, tumors were categorized into small or big lesions based on their diameter on histology sections (small <0.82 mm< big); this diameter was chosen because it was the median diameter of all tumors from a test KP cohort. To determine tumor weights, the biggest visible tumors per mouse at autopsy were macro-dissected and immediately weighed. To determine Glut1-3 correlation, KP tumor-bearing lung serial sections stained for Glut1 or Glut3 were analyzed.

## Preparation of lung tumors for electron microscopy (TEM and SEM) and NanoSIMS

Tumor-bearing KP and KPG1 mice at 21 weeks post-tumor initiation were injected intraperitoneally with a solution containing $^{13}$C-glucose (or $^{12}$C-glucose as control), four times every 45 min, beginning 3 hr prior to sacrifice, for a total of 7.5 g kg$^{-1}$. Mice were then perfused *via* the heart with a buffered mix of 1.0% glutaraldehyde and 2.0% paraformaldehyde in 0.1 M phosphate buffer (pH 7.4). After 2 hr, the lungs were removed and placed in the same fixative overnight. They were then sectioned with a vibratome (VT1200, Leica Microsystems) at a thickness of 100 μm, and the sections of interest, showing distinct tumors, washed thoroughly with cacodylate buffer (0.1 M, pH 7.4). These were then postfixed for 40 min in 1.0% osmium tetroxide with 1.5% potassium ferrocyanide in

cacodylate buffer, and then 40 min in 1.0% osmium tetroxide alone (in cacodylate buffer). They were finally stained for 30 min in 1% uranyl acetate in $H_2O$ before being dehydrated through increasing concentrations of EtOH and then embedded in Durcupan ACM (EMS, USA) resin. The resin-embedded sections were hardened for 24 hr in an oven at 65°C. Regions of interest were sectioned either at 50 nm thickness, and collected on single slot grids, for transmission electron microscopy (TEM), or at 0.5 µm thickness, and collected on silicon wafers, for scanning electron microscopy (SEM) and NanoSIMS analysis. Sections collected on grids were further contrasted with lead citrate and uranyl acetate solutions, and images taken using a transmission electron microscope (Spirit TEM, operating at 80 kV, FEI Company) with digital camera (Eagle CCD camera; FEI Company). SEM imaging of sections on silicon wafers for correlation with NanoSIMS images was performed using a Zeiss Gemini-SEM 500. Imaging was done at 3 kV, using an energy selective backscatter detector (EsB) with a grid bias of 1500 V. NanoSIMS images of cells selected from SEM images were acquired using a $Cs^+$ primary beam. Selected areas (usually 50 × 50 µm for entire cell images or 25 × 25 µm for specific areas within a cell) were first implanted in order to clean up the surface and reach stable emission conditions. The areas were then imaged by scanning the $Cs^+$-beam focused on the surface of the sample; beam diameter was about 200 µm and current 1.8 pA for the 50 × 50 µm images and about 100 µm with a current of 0.5 pA for the 25 × 25 µm images. Number of pixels were always 256 × 256 and dwell time 5 µs/pixel. For each image, eight layers were acquired and processed using LookatNanosims software (*Polerecky et al., 2012*). The layers were aligned and stacked. Using a SEM image aligned with the NanoSIMS images, areas of interests were drawn for every LBO.

## Protein extraction for western blot

For transient transfection, 293 T cells were co-transfected with an rtTA plasmid and the Lenti.*TRE*-PPARαΔ13_PGK-Cre construct using LF2000 (ThermoFisher Scientific), and were treated (or not) with 2 µg ml$^{-1}$ doxycycline one day later for 24 hr to monitor PPARαΔ13 induction. For all experiments, cells were lysed in RIPA (150 mM NaCl, 20 mM Tris pH 7.4, 0.1% SDS, 0.5% NP-40, 0.5% sodium deoxycholate, 1 mM sodium orthovanadate, one protease inhibitor cocktail tablet (11836145001, Roche)). Proteins were quantified using a BCA assay, and 50 µg (or 5 µg after Seahorse) was used for western blot. Antibodies used are anti-PPARα (PA1-822A, ThermoFisher Scientific, 1:1000; RRID:AB_2165595), anti-GLUT1 (07–1401, Millipore, 1:1000; RRID:AB_1587074), anti-GLUT3 (ab41525, Abcam, 1:1000; RRID:AB_732609), anti-β-tubulin (sc-9104, Santa Cruz Biotechnology, 1:1000; RRID:AB_2241191), and anti-γ-tubulin (T6557, Sigma-Aldrich, 1:10'000; RRID:AB_477584).

## Immunohistochemistry on mouse tissues

For histological analyses, lung lobes from tumor-bearing mice were fixed in 3,6% formaldehyde (VWR Chemicals, 20909.290) at 4°C for 24 hr, then rinsed once in PBS1x followed by a washing and a keeping in 70% absolute ethanol (VWR, C20820.362) before paraffin embedding. Lung paraffin blocks were sectioned at 4 µm using a rotary microtome (Thermo Scientific, HM325), mounted on slides, dried at 37°C overnight before being stored at 4°C. Lung tissue sections were dewaxed, followed by antigen retrieval in a 10 mM Trisodium Citrate buffer pH 6.0 for 20 min at 95°C. Slides were then washed three times for 5 min in PBS1x before endogenous peroxidase blockade with 0.3% hydrogen peroxide (VWR, 1.07209.0250) for 10 min at room temperature (RT). Next, lung sections were washed three times for 5 min in PBS1x + 0.2% Triton X-100 (Sigma Aldrich, 93420), and blocked in 5% goat serum in PBS1x + 0.2% Triton X-100 for 1 hr at RT. Mouse tissue sections were stained with anti-GLUT1 (ab115730, Abcam, 1:650; RRID:AB_10903230) or anti-GLUT3 (ab41525, Abcam, 1:200; RRID:AB_732609) antibodies in blocking solution, 5% goat serum in PBS1x + 0.2% Triton X-100, overnight at 4°C. The day after, slides were incubated 30 min at RT, and washed three times for 5 min in PBS1x + 0.2% Triton X-100. Tissue sections were incubated 1 hr in biotin-conjugated secondary antibodies in blocking solution, 5% goat serum in PBS1x + 0.2% Triton X-100. Sections were then washed three times for 5 min in PBS1x + 0.2% Triton X-100, incubated for 30 min in avidin-biotin HRP complexes (Vector Laboratories, Vectastain Elite ABC HRP kit, PK-6105), and revealed with a 3,3'-diaminobenzidine (DAB) peroxidase substrate kit (Vector Laboratories, SK-4100). Harris haematoxylin counterstains were performed. Sections immunohistochemically stained

were imaged with a Leica microscope (Olympus BX43/U-HGLGPS), or scanned using a slide scanner (Olympus, VS120-L100).

## Human microarray and immunohistochemistry

A tissue microarray (TMA) comprising 36 randomly selected human non-small cell lung cancers (18 stage IB-IIB LUAD, 18 stage IA-IIB LUSC based on the 7th Edition of the UICC TNM classification) was generated by SB using the Translational Research Unit Platform of the Institute of Pathology, University of Bern, as described previously (*Zlobec et al., 2013*). Tumors were resected between 1990 and 2007. Representative H&E slides from all cases were scanned using a digital slide scanner (Pannoramic P250, Budapest, Hungary). Using a digital TMA annotation tool, each scan was marked using eight cores per tumor of 0.6 mm diameter each. Annotated images were aligned with the tumor block and cored out automatically (TMA Grand Master, 3DHistech, Hungary), producing a next-generation Tissue Microarray (ngTMA). The ngTMA block was cut at 4 µm and a double IHC using anti-GLUT1 (AEC chromogen, red staining) and anti-HIF1α (DAB chromogen, brown staining) was performed. Data on HIF1α were not considered in this study. Anti-GLUT1 (07–1401, Millipore, 1:5000; RRID:AB_1587074) was incubated for 30 min. Antigen retrieval was done in citrate buffer for 30 min at 100℃. GLUT1 membrane staining on tumor cells was classified by an experienced pathologist (CN) into score 0 (no or weak staining in tumor cells in all eight cores), score 1 (intermediate staining in tumor cells in at least 1 of the eight cores) or score 2 (strong staining in tumor cells in at least 1 of the eight cores). No score 0 was present. Visual scoring was done using Scorenado, an automated core-wise digital image display and documentation tool based on QuPath elements (*Lytle et al., 2019*).

## RNA isolation, reverse transcription and real-time PCR

Single cell suspensions from macro-dissected tumors were obtained using a gentleMACS Octo Dissociator (Miltenyi Biotec). RNA was prepared for sequencing from bulk tumors (bulk Lenti) or from CD45⁻ (non-immune) tumor fractions (PGK-Cre, SPC-Cre and CC10-Cre). In the latter case, magnetic cell sorting to remove immune cells was performed using anti-CD45 MicroBeads (130-052-301, Miltenyi Biotec). RNA for real-time PCR or sequencing was extracted using TRIzol (15596018, Thermo-Fisher Scientific). For real-time PCR, 1 µg RNA was used for reverse transcription using High-Capacity cDNA Reverse Transcription Kit (4368814, ThermoFisher Scientific). Real-time PCR was done using Taqman universal PCR master mix (4324018, ThermoFisher Scientific) and Taqman probes for the following genes: *Slc2a1*: Mm00441480_m1; *Slc2a3*: Mm00441483_m1; *Slc2a2*: Mm00446229_m1; *Slc2a4*: Mm00436615_m1; *Slc2a5*: Mm00600311_m1; *Slc2a6*: Mm00554217_m1; *Slc2a7*: Mm01260610_m1; *Slc2a8*: Mm00444634_m1; *Slc2a9*: Mm00455122_m1; *Slc2a10*: Mm01249519_m1; *Slc2a12*: Mm02375931_s1; *Slc2a13*: Mm01306489_m1; *Slc5a2*: Mm00453831_m1; *Slc50a1*: Mm00485707_m1; gene expression level normalization gene *Rpl30*: Mm01611464_g1 (all from ThermoFisher Scientific, Catalog number: 4331182). Data were presented as $\Delta C_T$ values.

## Mouse mRNA sequencing, differential expression, and pathway analyses

Multiplexed libraries for mRNA-seq were prepared with the TruSeq stranded mRNA kit (Illumina) starting from 500 ng of good-quality total RNAs (RNA quality scores > 7 on the Fragment Analyzer). Sequencing was subsequently performed on a NextSeq 500 instrument (Illumina) on a high-output flow cell, yielding single-end reads of 85 nucleotides. Adapter sequences and low-quality ends were removed with Cutadapt (v1.9.1), trimming for TrueSeq and polyA sequences. Reads were aligned to mouse genome build mm10 using HISAT2 aligner (v2.0.3beta). Genes with low expression were filtered out (average transcripts per kilobase million < 20). Counts were normalized for library size using TMM method from EdgeR (v3.24.3) and voom from limma. Differential expression was computed with limma between Glut1-KO and WT control, or PPARαΔ13 and control.

We investigated pathways as defined in the Hallmark and KEGG collections of MSigDB (v6.0; *Subramanian et al., 2005*). Gene Set Enrichment Analysis (GSEA) was performed on genes ranked based on t-statistics obtained when comparing Glut1-KO and WT control, or PPARαΔ13 and control. Statistical significance was calculated by permutation tests (number of random permutations = $10^5$).

## Seahorse analyses

Single cell suspensions were prepared from macro-dissected tumors using gentleMACS Octo Dissociator (Miltenyi Biotec). Macro-dissected tumors were weighted and processed into DMEM, no glucose (11966025, ThermoFisher Scientific). Ten or five technical replicates per tumor (5 mg of tumor per replicate) were plated in a XF96 V3-PS cell culture microplate (Seahorse Bioscience, North Billerica) and the rest of cell suspensions were saved for real-time PCR. Cells were incubated during 2 hr in DMEM, no glucose at 37°C. Extracellular acidification rates (ECAR) were then measured using XFe96 Extracellular Flux Analyzer (Seahorse Bioscience, North Billerica, MA). ECAR measurements were performed at 37°C, with measurement cycles broken down into 2 min of mixing, 2 min of waiting and 3 min of data acquisition. During Seahorse assay, 10 mM of glucose was added, followed by 50 mM of 2-deoxyglucose (2-DG). ECAR measurements were reported as normalized to protein concentration. After Seahorse analysis, proteins were extracted from the cells for western blot analyses.

## Cell line authentication

Cell line authentication was performed by Microsynth AG. Profiling of the human cell lines was done using highly polymorphic short tandem repeat loci (STRs). STR loci were amplified using the PowerPlex 16 HS System (Promega). Fragment analysis was done on an ABI3730xl (Life Technologies) and the resulting data were analyzed with GeneMarker HID software (Softgenetics). A549, Calu-6, and H460 cell lines matched 100% to the DNA profile of A549 (ATCC CRM-CCL-185TM; RRID:CVCL_0023), Calu-6 (ATCC HTB-56TM; RRID:CVCL_0236), and NCI-H460 [H460] (ATCC HTB-177TM; RRID: CVCL_0459), respectively. SW 1573 cell line matched 93,3% to the DNA profile of the SW 1573 [SW-1573, SW1573] (ATCC CRL-2170TM; RRID:CVCL_1720).

## Mycoplasma test on human cell lines

Mycoplasma test (13100–01, Mycoplasma Detection Kit, SouthernBiotech) was performed on the 4 cell lines used for the in vitro experiments. All cell lines were mycoplasma negative.

## In vitro human non-small cell lung cancer experiments

Mycoplasma negative and authenticated human Calu-6, H460, A549 and SW 1573 cell lines were plated in 6-well culture plates in RPMI (Gibco, 21875–034) supplemented with 10% FBS (Gibco, 10270106) at $1.10^5$ cells per well. One day later, cells were transfected using lipofectamine RNAi max (Life Technologies, 13778150) with siRNAs: siCTR, si*SLC2A1*, si*SLC2A3*, or si*SLC2A1* + si*SLC2A3*. References and mixes of the siRNAs used for the different conditions are found in *Table 1*. 72 hr after transfection, cells were counted using Trypan blue dye exclusion assay (Invitrogen, T10282) and an automated cell counter (Invitrogen, Countess Automated Cell Counter). Cells were then lysed into TRIzol (Invitrogen, 15596018) to extract RNA.

## $^{18}$F-fluorodeoxyglucose with positron emission tomography ($^{18}$F-FDG PET)

In vivo measurements of the tumors' glucose uptake were performed on a group of animals (n = 14, i.e. 3 KP, 4 KPG1, 3 KPG3 and 4 KPG1G3, fasted overnight) with positron emission tomography (PET) as previously described in detail (*Lanz et al., 2014*). Mice were anesthetized with a mixture of 4% isoflurane (vol/vol) in 100% $O_2$ for 2 min and then maintained with a mixture of 1.5% isoflurane in 100% $O_2$ (0.9 L/min) during tail catheter insertion and initial glycaemia measurement. PET data were acquired on an avalanche photodiode–based LabPET-4 small-animal scanner (Gamma Medica, Sherbrook, Canada). Mice were prone positioned on a heat-regulated scanner bed with respiratory cushion and rectal temperature measurement probe, with the PET scanner field of view adjusted to cover the chest area. A bolus injection of $^{18}$F-fluorodeoxyglucose ($^{18}$F-FDG) (~20 MBq) was administered through the tail vein catheter within the first 30 s of the PET scan, followed by 150–200 µl of saline chase solution. During the 60 min scan, mice were maintained under 1.5–2% (vol/vol) isoflurane anesthesia in oxygen (0.9 L/min). Monitored temperature and breathing rate were maintained within a physiological range. Raw data were reconstructed with a maximum likelihood expectation maximization (MLEM) algorithm on a circular FOV of 60 mm of diameter and a standard voxel size of 0.5·0.5·1.18 mm, in three time frames of 20 min. The last time frame (labeling steady-state) was used

**Table 1.** References of the siRNAs used for in vitro experiments.

| siRNA | Mixes of | Species | Supplier | References |
|---|---|---|---|---|
| siCTR | siCTR | Human and Mouse | Invitrogen | Cat# 4390843 |
| si*SLC2A1* | si*SLC2A1*#1 | Human | ThermoFisher Scientific | s12926 Cat# 4390824 |
| | si*SLC2A1*#2 | Human | ThermoFisher Scientific | s12927 Cat# 4390824 |
| | siCTR | Human and Mouse | Invitrogen | 4390843 |
| si*SLC2A3* | si*SLC2A3*#1 | Human | ThermoFisher Scientific | s12931 Cat# 4390824 |
| | si*SLC2A3*#2 | Human | ThermoFisher Scientific | s12932 Cat# 4390824 |
| | siCTR | Human and Mouse | Invitrogen | Cat# 4390843 |
| si*SLC2A1* + si*SLC2A3* | si*SLC2A1*#1 | Human | ThermoFisher Scientific | s12926 Cat# 4390824 |
| | si*SLC2A1*#2 | Human | ThermoFisher Scientific | s12927 Cat# 4390824 |
| | si*SLC2A3*#1 | Human | ThermoFisher Scientific | s12931 Cat# 4390824 |
| | si*SLC2A3*#2 | Human | ThermoFisher Scientific | s12932 Cat# 4390824 |

to derive standardized uptake values (SUV) images and measure maximal SUV (SUV$_{max}$) values over tumor regions.

$$SUV\ [g/ml] = \frac{R_{ROI}[kBq/ml]}{A_{inj}[kBq]/w[g]}$$

where $R_{ROI}$ represents the radioactivity concentration measured in the region of interest (ROI), $A_{inj}$ is the injected activity corrected for the radiotracer decay and $w$ is the mouse weight.

PMOD 2.95 software (PMOD Technologies, Zurich) was used for the determination of the standardized uptake value (SUV) and for the delineation of the tumors' ROIs, based on a visual slice-by-slice matching with the corresponding μCT images. Tumors appearing on consecutive slices were grouped as single VOIs for the calculation of the SUVmax.

## Statistical analyses and graphic design

Statistical analyses were performed, and graphics produced using R version 3.5.1, Prism version eight or Microsoft Excel version 16.

## Acknowledgements

We thank the EPFL SV Histology, Gene Expression and BioEM Core Facilities for help with tissue sectioning, RNA sequencing and electron microscopy, respectively. We thank A Berns (NKI, Amsterdam) for allowing the use of SPC- and CC10-promoter Adeno-Cre vectors. We thank M Masin for initial *Slc2a1*$^{Flox}$ mouse work and G Boivin for help with μCT analyses. We thank L Michalik (UNIL, Lausanne) for providing us with the PPARαΔ13 construct, J Auwerx (EPFL, Lausanne) for sharing the Seahorse XF analyzer, M Eichmann and S Reinhard (Institute of Pathology, University of Bern) for developing the core-wise digital image display and documentation tool (Scorenado) and the Translational Research Unit, Institute of Pathology, University of Bern for technical expertise. We thank the CIBM team for the usage of CIBM resources.

# Additional information

## Funding

| Funder | Grant reference number | Author |
| --- | --- | --- |
| Swiss Cancer Research Foundation | KFS-3681-08-2015-R | Caroline Contat<br>Pierre-Benoit Ancey<br>Silvia Sabatino<br>Justine Pascual<br>Anouk Rossier<br>Etienne Meylan |
| Swiss National Science Foundation | PP00P3_133661 | Etienne Meylan |
| Swiss National Science Foundation | PP00P3_157527 | Caroline Contat<br>Etienne Meylan |
| Anna Fuller Fund | | Caroline Contat<br>Etienne Meylan |
| Emma Muschamp Foundation | | Caroline Contat<br>Pierre-Benoit Ancey<br>Silvia Sabatino<br>Etienne Meylan |

The funders had no role in study design, data collection and interpretation, or the decision to submit the work for publication.

## Author contributions

Caroline Contat, Conceptualization, Formal analysis, Supervision, Validation, Investigation, Visualization, Methodology, Project administration; Pierre-Benoit Ancey, Formal analysis, Validation, Investigation, Visualization, Project administration; Nadine Zangger, Software, Formal analysis, Validation, Investigation, Visualization; Silvia Sabatino, Justine Pascual, Louise Jensen, Anouk Rossier, Stéphanie Clerc-Rosset, Investigation; Stéphane Escrig, Formal analysis, Validation, Investigation, Visualization; Christine Goepfert, Formal analysis, Investigation; Bernard Lanz, Formal analysis, Investigation, Methodology; Mario Lepore, Investigation, Methodology; Rolf Gruetter, Inti Zlobec, Jeffrey C Rathmell, E Dale Abel, Resources; Sabina Berezowska, Anders Meibom, Resources, Supervision; Christina Neppl, Formal analysis, Investigation, Visualization; Graham William Knott, Formal analysis, Supervision; Etienne Meylan, Conceptualization, Formal analysis, Supervision, Funding acquisition, Validation, Visualization, Project administration

## Author ORCIDs

Graham William Knott (iD) http://orcid.org/0000-0002-2956-9052
E Dale Abel (iD) http://orcid.org/0000-0001-5290-0738
Etienne Meylan (iD) https://orcid.org/0000-0002-0899-2230

## Ethics

Human subjects: All human tissue experiments were performed with the permission of the Cantonal Ethics Commission of the Canton of Bern (KEK 200/14), which waived the requirement for written informed consent.
Animal experimentation: All mouse experiments were performed with the permission of the Veterinary Authority of the Canton de Vaud, Switzerland (Licences VD2391 and VD2663).

## Decision letter and Author response

Decision letter https://doi.org/10.7554/eLife.53618.sa1
Author response https://doi.org/10.7554/eLife.53618.sa2

## Additional files

### Supplementary files
• Transparent reporting form

### Data availability

The RNA sequencing data have been deposited to the GEO database (https://www.ncbi.nlm.nih.gov/geo/) and assigned the identifier: GSE138757.

The following dataset was generated:

| Author(s) | Year | Dataset title | Dataset URL | Database and Identifier |
|---|---|---|---|---|
| Contat C, Ancey PB, Zangger N, Sabatino S, Pascual J, Escrig Sp, Jensen L, Goepfert C, Lanz B, Lepore M, Gruetter R, Rossier A, Berezowska S, Neppl C, Zlobec I, Clerc-Rosset Sp, Knott GW, Rathmell JC, Abel ED, Meibom A, Meylan E | 2020 | Combined deletion of Glut1 and Glut3 impairs lung adenocarcinoma growth. | https://www.ncbi.nlm.nih.gov/geo/query/acc.cgi?acc=GSE138757 | NCBI Gene Expression Omnibus, GSE138757 |

The following previously published dataset was used:

| Author(s) | Year | Dataset title | Dataset URL | Database and Identifier |
|---|---|---|---|---|
| The International Cancer Genome Consortium (Hudson (Chairperson) | 2010 | International network of cancer genome projects | https://portal.gdc.cancer.gov/projects/TCGA-LUAD | The Cancer Genome Atlas, TCGA-LUAD |

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
