## [Decision Letter]

**Acceptance summary:**

This study uses conditional mice to show that GLUT1 inactivation is dispensable for tumor growth in a mouse model of lung cancer, while combined GLUT1/GLUT3 deletion leads to smaller tumors and longer survival. This supports a requirement for glucose uptake in lung cancer.

**Decision letter after peer review:**

Thank you for submitting your article "Combined deletion of Glut1 and Glut3 impairs lung adenocarcinoma growth" for consideration by *eLife*. Your article has been reviewed by three peer reviewers, and the evaluation has been overseen by a Reviewing Editor and Maureen Murphy as the Senior Editor. The following individuals involved in review of your submission have agreed to reveal their identity: Georgia Konstantinidou (Reviewer #2).

All reviewers appreciated the amount of work and potential importance, although also noted that some findings were expected or less novel and that further work would be needed to meet what is expected in *eLife*. After discussion amongst the reviewers, we would be willing to consider a substantially revised version that incorporates points raised during review. A list of essential revisions is included first, and then the full comments from the three reviewers. Please write a response to all reviewer points, but your revised manuscript should incorporate the essential revisions.

Essential revisions:

We appreciate the value in formally examining the requirement for GLUT1 and GLUT3 in lung cancers, however it was noted that glucose uptake was not formally examined in any system. This could be done using FDG-PET scan, and some data to assess how glucose uptake changes would strengthen the conclusions. We recognize that this may take longer than 2 months, but think it would greatly enhance the impact of the paper.

More evidence on how tumor histopathology is altered by GLUT1 and GLUT3 knockout should be included. For example, do double knockouts have no difference in adenoma growth but reduced progression to invasive cancers?

All appreciated is that cultured cells may be rewired by selection in high glucose media, and thus may not be the same as autochthonous tumors, but we agreed the comment regarding the human cancer cells lines from reviewer 2 should be fully addressed.

Reviewer #1:

The authors show that GLUT1 inactivation does not significant affect the growth of tumors in the KP GEM model of lung cancer, whereas combined GLUT1/GLUT3 deletion leads to smaller tumors and longer survival. The data are clear and are of potential relevance to therapy and understanding metabolic adaptations in lung cancer subsets. However, the findings are descriptive and provide only limited new information about cancer mechanisms. Moreover, the organization of the manuscript could be improved, with less attention to the negative results on PPARα and ultrastructural data of uncertain significance and additional characterization of the tumor phenotypes.

1) There are limitations in novelty and mechanistic insight. Prior reports (cited by the authors) anticipate some of the results-with the study by Scafoglio showing expression of GLUT1 as a late event in the KP model and in human specimens, and preferential early reliance on the Sglt2 transporter. Similarly data from the Hsieh study indicate that GLUT1 is dispensable for adenoma/adenocarcinoma development in the context of the KL model. The finding that co-deletion of GLU1 and GLUT3 does impact tumor growth in the KP model is interesting, but provides an overall incremental advance.

2) The relevance of the model to make predictions about glucose transport in human NSCLC is not certain. Although increasing GLUT3 levels are reported during tumor progression in patient samples, no functional data are shown using human-derived models. Moreover, relevance of GLUT1 vs GLUT3 in established tumors would be best addressed using acute inactivation methods.

3) More details should be provided in the analysis of the mouse models. Is the histology/grade affected by GLUT1 or GLUT1+GLUT3 inactivation?

Reviewer #2:

In this manuscript, Contat et al. proved that Glut1 deficiency fails to suppress tumorigenesis and they identified Glut3 as a compensatory mechanism. The combined deletion of Glut1 and Glut3 suppressed tumor growth, decreased tumor number and increased mouse survival.

Overall, the manuscript is well-written and the experiments are well performed.

Major Comments:

1) Goodwin et al., 2017, reported that GLUT1 is elevated at both the mRNA (TCGA) and protein levels in squamous lung cancer but is minimally expressed in lung adenocarcinomas. Instead, the authors here show similar high protein levels of GLUT1 in Figure 1B. This discrepancy should at least be discussed.

2) All the data in the manuscript are based on experimental evidence derived from mouse models. Although this is excellent, the authors can try to reproduce this synergistic action of GLUT1/3 inhibition in lung adenocarcinoma cell lines.

3) Figure 4. The suppression of tumor growth and tumor number is due to reduced cell proliferation or there`s also increased cell death?

Reviewer #3:

Contat et al. assess the role of Glut1 and Glut3 expression in the KP mouse model of lung cancer. They show that either transporter alone is dispensible for tumor growth, but deletion of both transporters extends mouse survival. Using nanoSIMS, they also show that glucose carbon accumulates in lamellar bodies, although what these structures are or how they relate to tumor growth is not clear. Overall, this work supports some requirement for glucose uptake in lung cancer and is consistent with another paper showing Glut1 is required for tumor growth in a different lung cancer model. The major deficiency of the study is that glucose uptake is never directly assessed in vivo. Some other points to consider are listed below.

1) The Abstract implies that glucose uptake may not be important based on Glut1 deletion, although then suggests it is important since Glut3 is required if Glut1 is absent. Also it is stated that expression of two glucose transporters is needed, but it also may be their function is redundant. While many would argue that a role for glucose metabolism in tumors is well established, some work has questioned this recently and being clear about their findings in the Abstract is important to prevent further confusion in the field.

2) The use of nanoSIMS to assess glucose fate following glucose transporter deletion is interesting, however this is a poor surrogate for glucose uptake (see below). Given that FDG-PET is a commonly used way to assess glucose uptake in patients and mice, one is left wondering what effect Glut1 loss has on overall glucose uptake. Can FDG-PET be compared in KP and KPG1 mice?

3) IP injection of 13C-labeled glucose will achieve a steady state labeling and thus differences in accumulation in lamellar bodies could be caused by differences in uptake between different models. This is another reason why assessment of FDG-PET uptake would be useful to interpret these data.

4) It is implied based on ex vivo data that some cells adapt to Glut1 loss via Glut3 expression while other rely on PPARα. This is speculative at best, particularly when the dominant negative construct had a marginal effect. A simpler experiment test might be to ask if PPARα expression anti-correlates with Glut3 expression via IHC in tumors?

5) For completeness the effect of Glut3 loss, and combined Glut1/Glut3 loss on FDG-PET uptake would improve the study.

6) As the authors point out, Glut1 is required for tumor growth in the KL model. The fact that KL tumors fail to express Glut3 is a nice explanation for why Glut1 is not needed in the KP model. If the authors also have data on Glut1 and Glut3 deletion in the KL model it would make this point more clearly and be nice to include, although if the data are not available it would take too long to generate these data for this paper.

7) The extension in mouse survival with dual deletion of Glut1 and Glut3 is fairly modest. Is their data on whether the tumors that eventually form still take up glucose via another mechanism?

---

## [Author Response]

Essential revisions:We appreciate the value in formally examining the requirement for GLUT1 and GLUT3 in lung cancers, however it was noted that glucose uptake was not formally examined in any system. This could be done using FDG-PET scan, and some data to assess how glucose uptake changes would strengthen the conclusions. We recognize that this may take longer than 2 months, but think it would greatly enhance the impact of the paper.

In response to this comment and to those of the reviewer #3, we have now used ^18^F-FDG PET to trace glucose uptake in KP, KPG1, KPG3 and KPG1G3 lung tumors. We have compared the maximum standardized uptake value (SUV_max_) of each tumor and observed that Glut1-deficient KP tumors incorporated significantly less ^18^F-FDG than the control lesions, which strengthened the NanoSIMS data (see manuscript Figure 2D). Glut3-deficient and control KP tumors presented a similar ^18^F-FDG absorption. This result might be explained by the fact that Glut3 positive KP tumors mainly co-express Glut1 and Glut3 (see manuscript Figure 4—figure supplement 1D), or that some small tumors do not yet express Glut3. Thus, in absence of Glut3, lesions might certainly incorporate ^18^F-FDG *via* Glut1. The ^18^F-FDG absorption of Glut1- and Glut3-deficient KP tumors was significantly reduced as compared to control KP lesions. Of note, the two KPG1G3 tumors showing a high ^18^F-FDG incorporation were probably escapers for one or both glucose transporters. These overall data suggest, as expected, that deleting glucose transporters in KP tumors reduces glucose uptake and that Glut1 mainly contributes to glucose uptake in KP tumors (Figure 4H-IA).

Material and methods:

^“18^F-fluorodeoxyglucosewith positron emission tomography (^18^F-FDG PET)

in vivo measurements of the tumors’ glucose uptake were performed on a group of animals (n = 14, i.e. 3 KP, 4 KPG1, 3 KPG3 and 4 KPG1G3, fasted overnight) with positron emission tomography (PET) as previously described in detail (Lanz et al., 2014). […] Tumors appearing on consecutive slices were grouped as single VOIs for the calculation of the SUVmax.”

More evidence on how tumor histopathology is altered by GLUT1 and GLUT3 knockout should be included. For example, do double knockouts have no difference in adenoma growth but reduced progression to invasive cancers?

To address this comment, we compared tumor grades by histopathology, based on a grading system established for the KP tumors (Jackson et al., 2005). Tumor comparisons were the following: KP vs. KPG1, KP vs. KPG3, and KP vs. KPG1G3.

Our results showed a significant decrease of high tumor grades (grades 4 and 5) in KPG1 vs. KP lesions and in KPG1G3 vs. KP tumors. We observed the same change when we grouped benign lesions (alveolar hyperplasias and tumor grades 1-3) and advanced lesions (grades 4-5). No differences in tumor grade distribution were seen between KP vs. KPG3 conditions.

These results are presented as new Figure 1G (KP vs. KPG1), new Figure 4—figure supplement 2D (KP vs. KPG3) and new Figure 4F (KP vs. KPG1G3).

Material and methods:

“Assessment of tumor parameters

Tumor grades were assessed from scanned haematoxylin and eosin (H&E) stained lung sections by a board- certified veterinary pathologist, Dr. med. vet., PhD, Goepfert Christine. This classification was based on a previous classification system (Jackson et al., 2005), and classified as follows: lesions were categorized as alveolar hyperplasia (AH), or as tumor on a 1-5 severity grading scale from grade 1, adenomas, to grade 5, adenocarcinomas.”

All appreciated is that cultured cells may be rewired by selection in high glucose media, and thus may not be the same as autochthonous tumors, but we agreed the comment regarding the human cancer cells lines from reviewer 2 should be fully addressed.

To address this comment, we used cell lines derived from human non-small cell lung cancer (NSCLC), grown in regular RPMI + 10% FBS (~11 mM glucose). Because our previous study highlighted that GLUT3 expression varies a lot between lung tumor cell lines (Masin et al., 2014), we selected four cell lines that express well GLUT3: A549, H460, Calu-6, and SW 1573. All cell lines were authenticated (Author response image 1) and mycoplasma negative (Author response image 1). We used trypan blue exclusion method to measure the number of viable cells 72 hours after transfection with control, GLUT1, GLUT3, or GLUT1 and GLUT3 siRNAs. The expression of GLUT1 and GLUT3 in these cell lines, as well as the efficiency of gene knockdown was assessed by real-time PCR. In all cell lines, the combination of GLUT1 and GLUT3 knockdown more strongly diminished tumor cell viability when compared to individual gene knockdown, which corroborates our in vivo data.

These results are presented as new Figure 4—figure supplement 3C-F.

Material and methods:

“Cell line authentication

Cell line authentication was performed by the Microsynth AG company. Profiling of the human cell lines was done using highly polymorphic short tandem repeat loci (STRs). STR loci were amplified using the PowerPlex 16 HS System (Promega). Fragment analysis was done on an ABI3730xl (Life Technologies) and the resulting data were analyzed with GeneMarker HID software (Softgenetics).”

Cell line authentication conclusions are detailed in Author response table 1.

**Author response table 1. resptable1:** Detailed conclusions of the cell line authentication.

Sample name	Organism	Conclusion
A549	Human	According to our analysis of the submitted sample there is no detectable contamination with human origin. The analyzed data of the submitted sample match 100% to the DNA profile of the cell line A549 (ATCC CRM-CCL^-^185TM) and 100% over all 15 autosomal STRs to Microsynth’s reference DNA profile of A549 (Mic_150733).
Calu-6	Human	According to our analysis of the submitted sample there is no detectable contamination with human origin. The analyzed data of the submitted sample match 100% to the DNA profile of the cell line Calu-6 (ATCC HTB-56TM) and 100% over all 15 autosomal STRs to the DNA profile of Calu-6 (Cellosaurus, RRID:CVCL_0236).
H460	Human	According to our analysis of the submitted sample there is no detectable contamination with human origin. The analyzed data of the submitted sample match 100% to the DNA profile of the cell line NCI-H460 [H460] (ATCC HTB-177TM) and 100% over all 15 autosomal STRs to the DNA profile of NCI-H460 (Cellosaurus, RRID:CVCL_0459).
SW 1573	Human	According to our analysis of the submitted sample there is no detectable contamination with human origin. The analyzed data of the submitted sample match 93.3% to the DNA profile of the cell line SW 1573 [SW-1573, SW1573] (ATCC CRL-2170TM) and 93.1% over all 15 autosomal STRs to the DNA profile of SW 1573 (Cellosaurus, RRID:CVCL_1720). Cell line samples matching at ≥ 80% of alleles across the eight core reference loci are said to be related.

**Author response image 1. respfig1:** All human NSCLC cell lines were mycoplasma negative. (**a**) PCR showing that Calu-6, H460, A549, and SW 1573 were tested negative for mycoplasma. Black arrow indicates the positive band for mycoplasma detection.

Mycoplasma test

Mycoplasma test (13100-01, Mycoplasma Detection Kit, Clone Set, Bioconcept) was performed on the 4 cell lines used for the in vitro experiments (Author response image 1).

In vitrohuman non-small cell lung cancer experiments

Mycoplasma negative and authenticated human Calu6, H460, A549 and SW 1573 cells were plated in 6-well culture plates in RPMI (Gibco, 21875-034) supplemented with 10% FBS (Gibco, 10270106) at 1.10^5^ cells per well. One day later, cells were transfected using lipofectamine RNAi max (Life Technologies, 13778150) with siRNAs: siCTR, siGLUT1, siGLUT3, or siGLUT1 + siGLUT3. References and mixes of the siRNAs used for the different conditions are found in Author response table 2. 72 hours after transfection, cells were counted using Trypan blue dye exclusion assay (Invitrogen, T10282) and an automated cell counter (Invitrogen, Countess Automated Cell Counter). Cells were then lysed into TRIzol (Invitrogen, 15596018) to extract RNA.

**Author response table 2. resptable2:** References of the siRNAs used for in vitro experiments.

siRNA	Mixes of	Species	Supplier	References
siCTR	siCTR	Human and Mouse	Invitrogen	4390843
siGLUT1	siGLUT1#1	Human	ThermoFisher Scientific	s12926
siGLUT1#2	Human	ThermoFisher Scientific	s12927
siCTR	Human and Mouse	Invitrogen	4390843
siGLUT3	siGLUT3#1	Human	ThermoFisher Scientific	s12931
siGLUT3#2	Human	ThermoFisher Scientific	s12932
siCTR	Human and Mouse	Invitrogen	4390843
siGLUT1 + siGLUT3	siGLUT1#1	Human	ThermoFisher Scientific	s12926
siGLUT1#2	Human	ThermoFisher Scientific	s12927
siGLUT3#1	Human	ThermoFisher Scientific	s12931
siGLUT3#2	Human	ThermoFisher Scientific	s12932

Reviewer #1:The authors show that GLUT1 inactivation does not significant affect the growth of tumors in the KP GEM model of lung cancer, whereas combined GLUT1/GLUT3 deletion leads to smaller tumors and longer survival. The data are clear and are of potential relevance to therapy and understanding metabolic adaptations in lung cancer subsets. However, the findings are descriptive and provide only limited new information about cancer mechanisms. Moreover, the organization of the manuscript could be improved, with less attention to the negative results on PPARα and ultrastructural data of uncertain significance and additional characterization of the tumor phenotypes.1) There are limitations in novelty and mechanistic insight. Prior reports (cited by the authors) anticipate some of the results-with the study by Scafoglio showing expression of GLUT1 as a late event in the KP model and in human specimens, and preferential early reliance on the Sglt2 transporter. Similarly data from the Hsieh study indicate that GLUT1 is dispensable for adenoma/adenocarcinoma development in the context of the KL model. The finding that co-deletion of GLU1 and GLUT3 does impact tumor growth in the KP model is interesting, but provides an overall incremental advance.

We agree that a lot remains to be discovered in terms of mechanistic insight, for example to understand the mechanisms of GLUT1 and GLUT3 co-regulation in lung cancer, which is a part that we have not addressed in the current study.

However, respectfully we do not agree that our study is limited in novelty. Glut1 and/or Glut3 gene deletion in the KP model of pure lung adenocarcinomas has never been reported before, and even if expression data show high Glut1 expression in advanced tumors this does not mean it is functionally implicated. The co-deletion data are, we believe, very important for drug design and potential clinical application. Indeed, our results suggest that GLUT1 or GLUT3 specific inhibitory compounds will not affect the development of lung adenocarcinoma, but that only dual inhibitors will.

Finally, thanks to this revision we increased our knowledge about in vivo tumor glucose uptake in the different genotypes (^18^F-FDG-PET imaging), and we report an unexpected accelerated KLG1 tumor growth compared to that of KL tumors (see our response to the reviewer #3 point 6), that we interpret as a sustained proliferative capacity consequent to the impaired LUAD-to-LUSC transdifferentiation caused by Glut1 deletion. This novelty was not reported in the Hsieh et al., 2019 study.

2) The relevance of the model to make predictions about glucose transport in human NSCLC is not certain. Although increasing GLUT3 levels are reported during tumor progression in patient samples, no functional data are shown using human-derived models. Moreover, relevance of GLUT1 vs GLUT3 in established tumors would be best addressed using acute inactivation methods.

First, to address this comment we have now used human tumor cell lines from non-small cell lung cancer (NSCLC) with GLUT1, GLUT3 or both knockdown, as described in Essential revisions point 3.

Second, to address the importance of glucose transporters in established tumors in vivo, we performed a pilot experiment using a second-generation KP mouse model, called KP^frt^ (*Kras^Frt-STOP-Frt-G12D/WT^; Tp53^Frt/Frt^*). We interbred KP^frt^ animals with Glut1^Flox/Flox^ mice, generating the KP^frt^G1 mouse model. We made a lentiviral vector expressing FlpO and CreER-T2 from a bi-directional promoter. We first validated, in vitro, the functionality of FlpO in recombining Frt sites, and CreER-T2 in recombining LoxP sites specifically upon 4-hydroxytamoxifen administration. We then intratracheally instilled KP^frt^G1 mice with the aforementioned lentiviral vector and followed tumor development using X-rays micro-computed tomography (μCT). At 41 weeks and 6 days post-tumor initiation, we injected mice with either vehicle control or with 4 doses of tamoxifen (100 μg/g of mouse every 3 days) (Author response image 2). We repeated tumor growth monitoring using μCT scans 2 weeks after the first tamoxifen injection, just before sacrifice. After sacrifice, we prepared lung sections for immunohistochemistry (IHC) staining.

To assess if Glut3 levels change upon acute Glut1 deletion, we analysed Glut1 and Glut3 protein expression by IHC staining. Our data showed that 50% of Glut1-deficient tumors (= tamoxifen-treated mice with no detectable expression of Glut1 in tumors) express Glut3 protein. In contrast, from this cohort 0% of Glut1-positive tumors (= vehicle-treated mice with detectable expression of Glut1 in tumors) express Glut3 protein (Author response image 2). Longitudinal μCT-based tumor growth rate measurements (n = 5 tumors from control-treated mice and n = 3 tumors from tamoxifen-treated mice) revealed that acute Glut1 deletion in established tumors seems to decrease tumor growth (Author response image 2). Besides, phospho-Histone H3 (pHH3) staining indicated that Glut1-deficient KP tumors have a significantly reduced proliferation as compared to control KP tumors (Author response image 2). Thus, these results suggest that acute Glut1 deletion in established KP tumors reduce tumor growth, which might be partially compensated by an increased Glut3 protein expression.

**Author response image 2. respfig2:** Acute Glut1 deletion in established KP^frt^ tumors may decrease tumor growth and may induce Glut3 expression. (**a**) Mouse model of acute Glut1 deletion. Flp-frt recombinations upon Flp-recombinase induce *Kras^G12D^* activation and *Tp53* deletion, driving tumor development. Intraperitoneal injection of tamoxifen during tumor development leads to Cre-recombinase activation and the subsequent Cre-lox recombination resulting in Glut1 deletion. (**b**) Glut1 and Glut3 immunohistochemistry (IHC) in KP^frt^ and KP^frt^G1. (**c**) Percent of Glut3 positive and negative tumors among (left panel) Glut1 proficient (KP) and (right panel) Glut1 deficient (KPG1) tumors. (**d**) Graph with mean ± s.d. displaying KP and KPG1 tumor fold change (n = 5 and 3 tumors, respectively) monitored during 56 days by X-rays micro-computed tomography (μCT), starting at 35 weeks and 5 days post-tumor initiation with tumor volumes set to 1. Blue arrows indicate the 4 tamoxifen injections leading to *Glut1* deletion. ns: not significant by Mann-Whitney test. (**e**) Dot plot with mean ± s.d. showing the percent of pHH3 positive cells per tumor between KP (n = 8) and KPG1 (n = 6) lesions. **: p < 0.01 by Mann-Whitney test.

However, this experiment is only a pilot study because the number of detectable tumors was small and the tumors took a very long time to grow. As a consequence, we are providing these data as part of this revision, but do not feel confident enough to add them as figure in the manuscript.

3) More details should be provided in the analysis of the mouse models. Is the histology/grade affected by GLUT1 or GLUT1+GLUT3 inactivation?

In response to this comment, we have now provided the grading-based histopathology assessment and comparison of KP, KPG1, KPG3 and KPG1G3 tumors. See the response to Essential revisions point 2.

Reviewer #2:In this manuscript, Contat et al. proved that Glut1 deficiency fails to suppress tumorigenesis and they identified Glut3 as a compensatory mechanism. The combined deletion of Glut1 and Glut3 suppressed tumor growth, decreased tumor number and increased mouse survival.Overall, the manuscript is well-written and the experiments are well performed.Major Comments:1) Goodwin et al., 2017, reported that GLUT1 is elevated at both the mRNA (TCGA) and protein levels in squamous lung cancer but is minimally expressed in lung adenocarcinomas. Instead, the authors here show similar high protein levels of GLUT1 in Figure 1B. This discrepancy should at least be discussed.

Our immunohistochemistry (IHC) in Figure 1B (see manuscript) shows that GLUT1 protein expression is either intermediate (score 1, 8/18 cases) or high (score 2, 10/18 cases) in the 18 samples of human LUAD we monitored, with no score 0 (no expression). In contrast, we agree with Goodwin et al., 2017, that GLUT1 is always highly expressed in LUSC (score 2 in 18/18 cases). We also agree with Goodwin et al., 2017, that GLUT1 mRNA expression is higher in LUSC than in LUAD, when analysing data from the TCGA database (Figure 1B of their paper and Author response image 3). However, we disagree that GLUT1 is minimally expressed in LUAD; GLUT1 is just less expressed in LUAD than in LUSC.

**Author response image 3. respfig3:** Glucose transporter expression in human lung squamous cell carcinomas (LUSC) and in human lung adenocarcinomas (LUAD). Gene expression level (RSEM) of glucose transporters in TCGA-LUSC (n = 501) and in TCGA-LUAD samples (n = 515).

Finally, we have compared the prognostic value of GLUT1 mRNA expression in LUAD and LUSC from the TCGA database. In LUAD, high GLUT1 expression is highly significantly associated with poor overall survival (see Figure 1—figure supplement 1A of our manuscript and the Kaplan-Meier analysis shown in the Author response image 4 hereafter).

In contrast, GLUT1 expression is not prognostic in LUSC (Kaplan-Meier analysis shown in the Author response image 4 hereafter). This is indeed a discrepancy with Goodwin et al. 2017, who reported an association between high GLUT1 and poor prognosis in LUSC. In their methods, they mentioned that they added 51 cases, which might make a difference. Nevertheless, GLUT1 expression is highly prognostic in LUAD, but not (according to TCGA) in LUSC, which is a strong argument to study GLUT1 in LUAD.

**Author response image 4. respfig4:** Survival analysis of 448 lung adenocarcinoma patients (TCGA-LUAD) and 475 lung squamous cell carcinoma patients (TCGA-LUSC). Median glucose transporter GLUT1 expression value splits down the patient tumors into low and high GLUT1 expressers. Kaplan-Meier plots show that GLUT1 expression is associated with a poor overall survival in patients with (a) LUAD tumors, whereas it is not prognostic in patients with (b) LUSC lesions. p-values were computed with Wald test.

2) All the data in the manuscript are based on experimental evidence derived from mouse models. Although this is excellent, the authors can try to reproduce this synergistic action of GLUT1/3 inhibition in lung adenocarcinoma cell lines.

We thank the reviewer for highlighting the value of autochthonous mouse models of lung cancer. In response to this comment, we have now used human tumor cell lines from non-small cell lung cancer with GLUT1, GLUT3 or both knockdown, as described in Essential revisions point 3.

3) Figure 4. The suppression of tumor growth and tumor number is due to reduced cell proliferation or there`s also increased cell death?

We have now looked at cell proliferation and cell death by IHC using anti-phospho-Histone H3 (pHH3), and anti-cleaved caspase-3 (CC3) staining, respectively. Unfortunately, cleaved caspase-3 staining did not reveal any apoptotic cell, which may be explained by the fact that we focused on a quick apoptotic process, the cleaved caspase-3 detection, at a specific time windows, the sacrifice. We may consequently have missed this rapid event on the IHC staining. Concerning cell proliferation, we surprisingly quantified an increased proliferation rate of KPG1G3 tumors compared to control KP lesions (Author response image 5), which was particularly evident in small lesions. We then hypothesised, but unfortunately were not able to test it, that the immune microenvironment of KPG1G3 tumors might be more active than the one of the KP lesions. Ping-Chih Ho et al., 2015, indeed demonstrated that the anti-tumor T cell effector functions were regulated by glucose availability in tumor microenvironment with a glucose-deficient microenvironment suppressing the T cell anti-tumor response. KPG1G3 tumors absorbing much less glucose than the control KP lesions (see Essential revisions point 1), glucose might be available at higher concentration in the microenvironment, allowing the anti-tumor T cell effector functions to normally behave and attack the tumor cells. KPG1G3 tumor development should thus be controlled by the immune cells. The communication between glucose uptake in tumor cells and the functionality of the tumor-associated immune cells is an area of interest of our laboratory, but we believe these studies will go beyond the scope of the present manuscript.

**Author response image 5. respfig5:** KPG1G3 tumors proliferate significantly more than KP lesions. (**a**) Dot plot with mean ± s.d shows percent of pHH3 positive cells per KP or KPG1G3 tumors (n = 4, 6 mice and n = 115, 101 tumors, respectively) at sacrifice. ****: p < 0.0001 by Mann-Whitney test.

Reviewer #3:Contat et al. assess the role of Glut1 and Glut3 expression in the KP mouse model of lung cancer. They show that either transporter alone is dispensible for tumor growth, but deletion of both transporters extends mouse survival. Using nanoSIMS, they also show that glucose carbon accumulates in lamellar bodies, although what these structures are or how they relate to tumor growth is not clear. Overall, this work supports some requirement for glucose uptake in lung cancer and is consistent with another paper showing Glut1 is required for tumor growth in a different lung cancer model. The major deficiency of the study is that glucose uptake is never directly assessed in vivo. Some other points to consider are listed below.1) The Abstract implies that glucose uptake may not be important based on Glut1 deletion, although then suggests it is important since Glut3 is required if Glut1 is absent. Also it is stated that expression of two glucose transporters is needed, but it also may be their function is redundant. While many would argue that a role for glucose metabolism in tumors is well established, some work has questioned this recently and being clear about their findings in the Abstract is important to prevent further confusion in the field.

We thank the reviewer for this recommendation, and in response we have modified the Abstract. We hope this version is now clearer about our findings.

2) The use of nanoSIMS to assess glucose fate following glucose transporter deletion is interesting, however this is a poor surrogate for glucose uptake (see below). Given that FDG-PET is a commonly used way to assess glucose uptake in patients and mice, one is left wondering what effect Glut1 loss has on overall glucose uptake. Can FDG-PET be compared in KP and KPG1 mice?

To respond to this critical point, raised as Essential revision point 1, we have now used ^18^F-FDG-PET imaging and have compared tumors from KP, KPG1, KPG3 and KPG1G3 mice. See the response to Essential revisions point 1.

3) IP injection of 13C-labeled glucose will achieve a steady state labeling and thus differences in accumulation in lamellar bodies could be caused by differences in uptake between different models. This is another reason why assessment of FDG-PET uptake would be useful to interpret these data.

We thank the reviewer for this comment and have now performed ^18^F-FDG-PET imaging. See the response to Essential revisions point 1.

4) It is implied based on ex vivo data that some cells adapt to Glut1 loss via Glut3 expression while other rely on PPARα. This is speculative at best, particularly when the dominant negative construct had a marginal effect. A simpler experiment test might be to ask if PPARα expression anti-correlates with Glut3 expression via IHC in tumors?

To answer to this interrogation, we stained serial KP and KPG1 tumor-bearing lung sections for Glut3 and Pparα by immunohistochemistry (IHC) (Author response image 6). We then quantified the percent of Glut3 or Pparα positive cells per tumor and correlated these data for the KP and KPG1 lesions. We observed that Glut3 and Pparα expressions correlate neither in KP nor in KPG1 tumors (Author response image 6).

**Author response image 6. respfig6:** Glut3 and Pparα expressions correlate neither in KP nor in KPG1 tumors. (a) Representative serial KPG1 tumor-bearing lung sections stained by immunohistochemistry (IHC) for (upper panel) Glut3 and (lower panel) Pparα. Tumors analyzed for this KPG1 mouse are encircled in red. Scale bars: 2 mm. (**b-c**) Graphs showing the Pearson correlation between Glut3 and Pparα expressions determined by IHC staining in (b) KP (n = 45) and (c) KPG1 (n = 28) tumors. Line of identify, R squared are indicated on graphs. ns: not significant.

5) For completeness the effect of Glut3 loss, and combined Glut1/Glut3 loss on FDG-PET uptake would improve the study.

We have now used ^18^F-FDG-PET imaging comparing all genotypes, which means that we analysed KP, KPG1, KPG3, KPG1G3 tumors. See the response to Essential revisions point 1.

6) As the authors point out, Glut1 is required for tumor growth in the KL model. The fact that KL tumors fail to express Glut3 is a nice explanation for why Glut1 is not needed in the KP model. If the authors also have data on Glut1 and Glut3 deletion in the KL model it would make this point more clearly and be nice to include, although if the data are not available it would take too long to generate these data for this paper.

We thank the reviewer for raising this point.

Because we measured only very low Glut3 mRNA expression and failed to detect the protein in KL tumors (see manuscript Figure 4—figure supplement 1A-B), we never generated KLG3 mice.

We had generated KLG1 mice and studied KL and KLG1 tumor development. However, we did not include these data in the manuscript since analysis of tumors from KLG1 mice have been reported recently (Hsieh et al., 2019). In that paper, Hsieh et al., 2019 showed a near-absence of lung squamous cell carcinoma (LUSC) in Glut1-deficient condition and no difference in total tumor burden between KL and KLG1 tumors. They concluded that Glut1 is critical for adenocarcinoma transdifferentiation and so LUSC development. Of note, in this model LUSCs develop from a transdifferentiation of LUADs (Han et al., 2014).

Accordingly to Hsieh et al., 2019 study, we confirmed that Glut1 deletion in KL tumors impacts neither overall mouse survival, nor tumor initiation (Author response image 7). Unexpectedly, after having used X-rays micro-computed tomography (μCT) analysis to longitudinally follow KLG1 and KL tumor development, we observed that Glut1 deficiency led to a significantly accelerated tumor growth rate (Author response image 7). Intrigued by these findings, we collected a few tumors from both genotypes after sacrifice and performed an RNA sequencing. Our analyses identified an increased cell cycle signature in Glut1-deficient KL lesions (Author response image 7), which corroborates the observed increased tumor growth of KLG1 lesions. We interpret these data as follows: KL tumors require Glut1 for LUAD-to-LUSC transdifferentiation, and Glut1 deletion prevents this, as reported by Hsieh et al., 2019. However, in Glut1 wild-type tumors, this transdifferentiation happens at the expense of a decreased tumor cell division. Accordingly, our data demonstrate that tumor cell proliferation marks are more elevated in absence of Glut1 and that in vivo tumor growth is faster: KLG1 LUAD tumors keep growing. Although we present these data for the reviewers, we do not feel confident enough yet to include them in the present manuscript, as we need confirmation of these surprising findings before.

**Author response image 7. respfig7:** The increased KLG1 tumor growth does not impact the overall mouse survival.

(a) Kaplan-Meier curves of KL (n = 6) and KLG1 (n = 5) mice. ns: not significant by Log-rank test. (b) Dot plot with mean ± s.e.m. displaying the average KL and KLG1 tumor number per mouse (n = 9 and 6, respectively). ns: not significant by Mann-Whitney test. (c) Graph with mean ± s.e.m. showing KL and KLG1 tumor fold changes (n = 10 and 8, respectively) monitored during 62 days by X-rays micro-computed tomography (μCT), starting at 17 weeks and 3 days post-tumor initiation with tumor volumes set to 1. ns: not significant by Mann-Whitney test; *: p < 0.05 by Mann-Whitney test. (d) Heatmap showing expression of the G2/M checkpoint genes from the Hallmark gene set in KL (n = 4) versus KLG1 (n = 6) tumors. p-value: 2.10^-3^ by permutation test. Color legend: black: median expression, red: above median, green: below median for each gene./Author response image 7 title/legend>

7) The extension in mouse survival with dual deletion of Glut1 and Glut3 is fairly modest. Is their data on whether the tumors that eventually form still take up glucose via another mechanism?

The extension in mouse survival corresponds to a median survival of 172 days and 211.5 days post-tumor initiation in KP and KPG1G3 mice, respectively. In other words, KPG1G3 mice lifespan is about 23% longer than the one of the control mice. Of note, in this survival analysis 2 out of 6 KPG1G3 mice analyzed were still alive and looked fine when we stopped the experiment.

Based on our new ^18^F-FDG-PET imaging of glucose uptake (See the response to Essential revisions point 1), we can now conclude that although KPG1G3 tumors still absorb glucose, this uptake is significantly decreased as compared to control KP lesions. Thus, to determine if Glut1- and Glut3-deficient tumors might incorporate this remaining glucose *via* another glucose transporter, we performed a gene expression analysis of glucose transporter family members on KP and KPG1G3 tumors. This analysis revealed that no other glucose transporter is induced upon Glut1 and Glut3 deletion in KP tumors. These data could then lead to the hypothesis that at least a part of the remaining ^18^F-FDG-PET signal detected in KPG1G3 tumors comes from glucose absorbed by the cells of the tumor immune microenvironment.

Of note, we could not detect any Sglt2 mRNA expression at all in KP and KPG1G3 samples, Glut5 mRNA expression was undetermined in KPG1G3 tumors, and only one KPG1G3 lesion had a detectable Glut7 mRNA expression explaining why we could not have a statistical test between the KP and KPG1G3 conditions for these glucose transporters.

Although we analysed the same KP and KPG1G3 tumors, we evaluated fewer tumors (n = 7 KP and n = 3 KPG1G3) for the expressions of Glut2, Glut3, Glut4, Glut5, Glut6, Glut7, Glut8, Glut9, Glut10, Glut12, Glut13, Sweet and Sglt2, than for the expression of Glut1 and Glut3 (n = 8 KP and n = 10 KPG1G3). We indeed did not have enough cDNA for each tumor to perform all these experiments.

These results are presented as new Figure 4 —figure supplement 4A.